# EEG-Language Pretraining for
# Highly Label-Efficient Clinical Phenotyping

**Sam Gijsen** [1 2]  **Kerstin Ritter** [1 2]

## Abstract

Multimodal language modeling has enabled breakthroughs for representation learning, yet remains unexplored in the realm of functional brain data for clinical phenotyping. This paper pioneers EEG-language models (ELMs) trained on clinical reports and 15000 EEGs. We propose to combine multimodal alignment in this novel domain with timeseries cropping and text segmentation, enabling an extension based on multiple instance learning to alleviate misalignment between irrelevant EEG or text segments. Our multimodal models significantly improve over EEG-only models across four clinical evaluations and for the first time enable zero-shot classification as well as retrieval of both neural signals and reports. In sum, these results highlight the potential of ELMs, representing significant progress for clinical applications.

Medical neuroimaging, including electroencephalography (EEG), has lagged behind other fields in leveraging the significant advancements of deep learning. While EEG sees widespread clinical use for clinical phenotyping such as pathology detection, in particular for epilepsy (Binnie & Stefan, 1999; Jing et al., 2020) as well as sleep disorders (Malhotra & Avidan, 2013), available annotated data is scarce. As the impressive scaling properties of deep learning are now well described (Kaplan et al., 2020; Smith et al., 2023), self-supervised learning (SSL) is a promising direction by enabling pretraining with unlabeled data and thereby increasing available training sample sizes (Hadsell et al., 2006; Chen et al., 2020). Various such approaches have shown initial success when applied to EEG. These include methods relying on data-augmentations (Mohsenvand et al.,

2020; Yang et al., 2021), the temporal ordering of EEG data (Banville et al., 2021), subject identities of EEG crops (Gijsen & Ritter, 2025), as well as masking and reconstruction (Jiang et al., 2024). However, these are hindered by the difficulty of creating appropriate data augmentations and, especially reconstruction techniques, by low signal-to-noise. Thus, progress in the medical context has lagged, likely further exacerbated by the modality displaying high similarity between pathologies.

Meanwhile, important further progress was made in computer vision by leveraging natural language as a signal during pretraining (Radford et al., 2021). Specifically, contrastive approaches which aim to align embeddings of image-text pairs have shown to yield representations powerful for downstream tasks in radiology (Zhang et al., 2022a; 2023). Given that success in radiology is also believed to be bottlenecked by the availability of labeled data and the reliance on fine-grained information (Zhang et al., 2022a), this joint modeling approach is a particularly interesting and novel application for the challenging problem of medical EEG. Fortunately, this is made possible by the clinical reports of physicians which accompany hospital EEG recordings and contain information about the patient and recording itself (Obeid & Picone, 2016).

However, language-EEG pretraining also entails unique challenges. First, datasets are generally smaller than those used in radiology and especially computer vision. Second, the clinical reports tend to be highly heterogeneous. While previous applications have paired natural and medical images with short captions (Radford et al., 2021; Zhang et al., 2022a), EEG reports tend to span multiple paragraphs and include information irrelevant to downstream clinical tasks, potentially hindering the pretraining process. Moreover, they do not contain temporal information about when events occurred during the recording.

The current work presents the application of aligning functional brain data with medical textual information for the first time by training EEG-language models (ELMs). To overcome the challenging formats of modalities, constituting long timeseries and multiparagraph reports, we propose sub-unit alignment. To address inconsistent relevance of EEG-text pairs, we additionally propose an extension draw-

[1]Charité – Universitätsmedizin Berlin, Department of Psychiatry and Psychotherapy, Berlin, Germany [2]Hertie Institute for AI in Brain Health, University of Tübingen, Germany. Correspondence to: Sam Gijsen <sam.gijsen@charite.de>.

*Proceedings of the 42nd International Conference on Machine Learning*, Vancouver, Canada. PMLR 267, 2025. Copyright 2025 by the author(s).

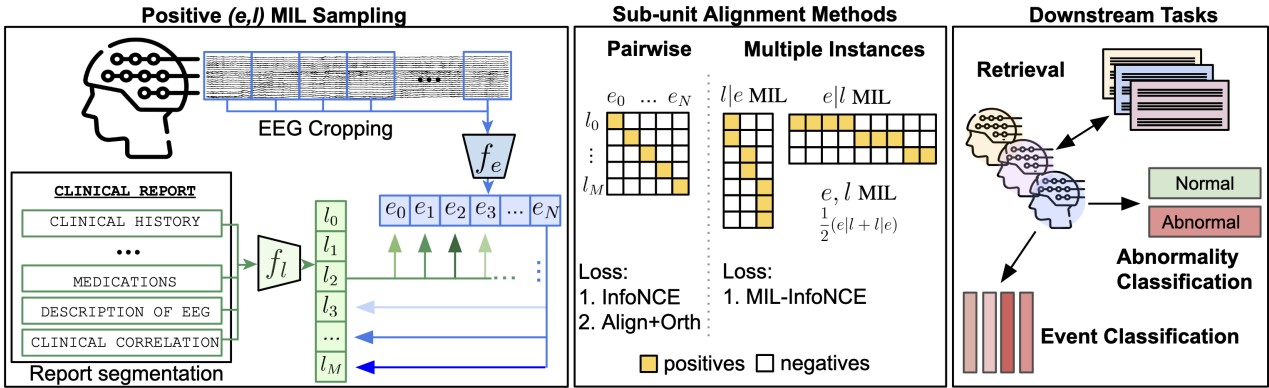

*Figure 1.* Overview of the methodology. (Left) The ELM-MIL approach allows flexible multimodal alignment by cropping EEG and segmenting medical reports. We sample multiple positives in a cross-modal fashion, such that each EEG crop can be aligned with any number of segments from the paired report ($l|e$). Vice versa, text can be aligned selectively to crops across the EEG recording ($e|l$), illustrated by the differently shaded arrows. (Middle) An overview of investigated methods by visualizing the cross-modal similarity matrices. (Right) To evaluate models, we perform bidirectional retrieval analyses and perform multiple pathology detection tasks.

ing on insights from the field of multiple instance learning (MIL). Furthermore, we investigate how to best handle the heterogeneity of medical EEG reports. Specifically, we filter reports and perform content-based text segmentation, enabling inference on the relative importance of the different information sources. Our approach allows us to provide the first evidence of considerable retrieval capabilities for clinical reports and EEG. We furthermore test downstream performance of ELMs on classifying pathological EEG, which is a widespread clinical task, using four evaluations. These tests include zero-shot classification, leveraging language capabilities to show our approach's flexibility. Our results constitute considerable increases in pathology detection performance, especially in scenarios with few labels. These are particularly relevant for clinical contexts, which tend to operate with smaller datasets compared to many common areas of deep learning applications.

## 1. Related work

- **Self-supervised learning with EEG data.** SSL with EEG data has been predominantly applied to emotion recognition (Zhang et al., 2022b; Wang et al., 2023), motor imagery (Cheng et al., 2020; Rommel et al., 2022), sleep staging (Yang et al., 2021; Rommel et al., 2022), as well as pathology detection. For the latter application, the temporal order of EEG crops was used initially to demonstrate label-efficient representation learning (Banville et al., 2021). Augmentation-based contrastive learning, combined with larger EEG encoders trained on multiple datasets, further improved pathology detection (Mohsenvand et al., 2020). Still, using subject identities as proxy labels during pretraining was found to improve disease detection over augmentation-based methods (Gijsen & Ritter, 2025). Recent studies have explored

the use of transformers (Yang et al., 2024; Jiang et al., 2024), with a focus on scaling while adopting tokenization in an attempt to improve the challenge of effective cross-dataset EEG training.

- **Using EEG for pathology detection.** While SSL shows good performance for pathology detection, it is particularly in contexts with little annotated data that it performs well. When more labeled data is available, expert-based feature extraction combined with traditional machine learning classifiers are competitive together with supervised deep learning (Roy et al., 2019; Gemein et al., 2020; Western et al., 2021; Kiessner et al., 2023; Darvishi-Bayazi et al., 2024). This trend has also been observed in other EEG applications (Schirrmeister et al., 2017; Lotte et al., 2018). This may indicate that inter-rater variability in EEG classification may create a performance ceiling (Engemann et al., 2018; Gemein et al., 2020), highlighting the importance of improving classification with limited labels.

- **Medical multimodal language modeling.** Medical vision-language modeling aims to guide self-supervised pretraining on medical images using textual information in reports, with performance on a variety of downstream tasks benefiting as a result (Huang et al., 2021; Wang et al., 2022; Zhang et al., 2022a). Due to less available data in the medical domain, using a pretrained, frozen language encoder was found to boost downstream performance while considerably reducing computational cost (Liu et al., 2023a). Nevertheless, this line of work has focused mainly on the ECG, X-ray, CT images, and structural MRI images (Chen et al., 2023; Lalam et al., 2023; Liu et al., 2023b). Recent advances outside the medical domain include multi-task strategies, both during pretrain-

ing by integrating contrastive learning and self-supervised losses (Tang et al., 2025; Tschannen et al., 2025), as well as finetuning on multiple downstream tasks (Dai et al., 2023; Liu et al., 2024). Further exploration involves moving compute from unimodal encoding to multimodal fusion (Kim et al., 2021).

- **Multiple instance learning.** MIL has seen only limited exploration for EEG. Initial studies have investigated the framework by casting crops of EEG as instances and training classifiers for emotion recognition (Caicedo-Acosta et al., 2019), motor imagery (Collazos-Huertas et al., 2020), mental disorders (Sadatnejad et al., 2019), and sleep apnea (Sadatnejad et al., 2019). Of these, only the latter has relied on deep learning.

## 2. Methods

### 2.1. Pretraining

#### 2.1.1. EEG-LANGUAGE PRETRAINING

Here we detail the setup for pretraining ELMs. Whereas vision-language models are typically trained by aligning a 2D image with a short caption (Radford et al., 2021; Zhang et al., 2022a), EEG-language modeling is confronted with long EEG time series and multi-paragraph medical reports. To overcome this, we employ text segmentation and time series cropping to create multiple non-overlapping samples per modality and subject. Next, we propose sub-unit alignment by pretraining on these cropped samples. In addition to considerably increasing sample size, this enables the extension of successful approaches in vision-language models. We initially describe two strategies for sub-unit alignment. First, EEG and text representations may be projected using neural networks to a new, shared latent space prior to alignment (as in CLIP; (Radford et al., 2021; Zhang et al., 2022a)), denoted henceforth as $\text{ELM}_{e,l}$. Alternatively, the EEG embeddings may be projected into the output space of the language model (as in M-FLAG by Liu et al. (2023a)), denoted as $\text{ELM}_l$ and trained using a bespoke loss function. This approach was found to reduce latent collapse in smaller data settings (Liu et al., 2023a). Following a description of these models, we will introduce an extension based on MIL.

For EEG-language pretraining we assume the paired input $(\mathbf{x}_{e,i}, \mathbf{x}_{l,i})$. Here $\mathbf{x}_{e,i} \in \mathbb{R}^{c \times s}$ denotes one or a batch of crops of EEG signal with $c$ channels and $s$ time samples belonging to EEG recording $i$. Meanwhile, neural signals of recording $i$ as well as patient information is described in $\mathbf{x}_{l,i}$, which represents a natural language text report. The main goal is to train the EEG encoder function $f_e$, which projects a crop of EEG signal into a vector of lower dimensionality. Following pretraining, this encoder function $f_e$ can be used for downstream applications such as pathology detection.

Dropping the recording subscript $i$ for brevity, each pair

$(\mathbf{x}_e, \mathbf{x}_l)$ is projected into the vectors $\mathbf{e} \in \mathbb{R}^d$ and $\mathbf{l} \in \mathbb{R}^d$ respectively. For every $\mathbf{x}_e$, text of the associated report is sampled according to $\tilde{\mathbf{x}}_l = z_l(\mathbf{x}_l)$, where $z_l$ represents the language sampling function detailed below. First, both the EEG crop $\mathbf{x}_e$ and text $\tilde{\mathbf{x}}_l$ are encoded into vectors $\mathbf{h}_e$ and $\mathbf{h}_l$. For $\text{ELM}_{e,l}$, we use projectors $g_e$ and $g_l$ to yield vectors $\mathbf{e}$ and $\mathbf{l}$, whereas for $\text{ELM}_l$ the text embeddings are not projected:

$$\mathbf{e} = g_e(f_e(\mathbf{x}_e)) \tag{1}$$

$$\mathbf{l} = \begin{cases} g_l(f_l(\tilde{\mathbf{x}}_l)) & \text{if } \text{ELM}_{e,l} \\ f_l(\tilde{\mathbf{x}}_l) & \text{if } \text{ELM}_l \end{cases} \tag{2}$$

To enable multimodal pretraining, the projectors $g_e$ and $g_l$ map $\mathbf{e}$ and $\mathbf{l}$ to a shared latent space with identical dimensionality $d$. For $\text{ELM}_l$, this is achieved by having $g_e$ project to the native dimensionality of the text encoder $f_l$.

As paired medical EEG data and clinical reports are scarce, training the text encoder function $f_l$ from scratch is unlikely to be successful. Furthermore, employing an existing language model and finetuning the model during multimodal pretraining can lead to training instability and collapse of the latent space (Jing et al., 2021; Liu et al., 2023a). To prevent resulting information loss, we follow the recommendations by Liu et al. (2023a) to use a pretrained language model for $f_l$ and freeze its weights during training. For $\text{ELM}_l$, we adopt their proposed composite loss to learn $f_e$ and $g_e$:

$$\mathcal{L}_{total} = \mathcal{L}_{align} + \mathcal{L}_{orth} \tag{3}$$

$$\mathcal{L}_{align} = \|\hat{\mathbf{e}} - \hat{\mathbf{l}}\|_2^2 = 2 - 2\hat{\mathbf{e}}^\top \hat{\mathbf{l}} \tag{4}$$

$$\mathcal{L}_{orth} = \sum_{j=1} \left(1 - \left(\hat{\mathbf{h}}_e^\top \cdot \hat{\mathbf{h}}_e\right)_{jj}\right)^2 \tag{5}$$

$$+ \sum_{j \neq k} \left(\hat{\mathbf{h}}_e^\top \cdot \hat{\mathbf{h}}_e\right)_{jk}^2 \tag{6}$$

where $\{j, k\} \in \{1, ..., \dim\left(\hat{\mathbf{h}}_e\right)\}^2$, $\hat{\mathbf{h}}_e$ denotes a batch of $\ell_2$-normalized EEG embeddings prior to projection and $\hat{\mathbf{e}}$ denotes the normalized projected embeddings. Text embeddings $\hat{\mathbf{l}}$ are likewise normalized. Whereas $\mathcal{L}_{align}$ minimizes the difference between $\hat{\mathbf{e}}$ and $\hat{\mathbf{l}}$, $\mathcal{L}_{orth}$ promotes independence between latent dimensions of $\hat{\mathbf{h}}_e$. More specifically, the latter is achieved by manipulating the empirical correlation matrix, where the diagonal and off-diagonal elements are pushed to 1 and 0 respectively (Liu et al., 2023a).

Meanwhile, $\text{ELM}_{e,l}$ relies on the cosine similarities between normalized EEG and text embeddings, $s_{j,j}^{e2l} = \hat{\mathbf{e}}_j^\top \hat{\mathbf{l}}_j$, and between text and EEG, $s_{j,j}^{l2e} = \hat{\mathbf{l}}_j^\top \hat{\mathbf{e}}_j$, with $j = 1, 2, 3, ..., B$ for batch size $B$ (Radford et al., 2021). The multimodal contrastive InfoNCE loss uses a temperature hyperparameter

$\tau$ and is formulated as:

$$\mathcal{L}_{j,k}^{e2l} = -\log \frac{\exp\left(s_{j,k}^{e2l}/\tau\right)}{\sum_{m=1}^{B} \exp\left(s_{j,m}^{e2l}/\tau\right)} \quad (7)$$

$$\mathcal{L}_{j,k}^{l2e} = -\log \frac{\exp\left(s_{j,k}^{l2e}/\tau\right)}{\sum_{m=1}^{B} \exp\left(s_{j,m}^{l2e}/\tau\right)} \quad (8)$$

$$\mathcal{L}_{align} = \frac{1}{2B} \sum_{j=1}^{B} \sum_{k=1}^{B} \left(\mathcal{L}_{j,k}^{e2l} + \mathcal{L}_{j,k}^{l2e}\right) \quad (9)$$

**Multiple instance learning.** While previous approaches aim to align text and EEG crops uniformly, certain text segments likely describe specific EEG sections more accurately than others. Therefore, we introduce a MIL alignment strategy that builds on $ELM_{e,l}$ and accommodates multiple positive samples, allowing for more nuanced multimodal relationships. Whereas MIL approaches often rely on operations such as max-pooling to focus on single positive samples, we rely on insights from the video-text alignment approach (MIL-NCE) by Miech et al. (2020). For a given text sample $\mathbf{x}_l$, we sample multiple positive EEG crops $\mathbf{x}_e$ from the paired recording to approximate the $P(e|l)$ distribution, while for an EEG crop, multiple text segments are sampled to model the $P(l|e)$ distribution. We combine these and sample positives for each EEG and text report respectively to approximate $P(e,l)$ via bidirectional alignment. This approach effectively relaxes the assumption of strong alignment for each individual $(\mathbf{x}_e, \mathbf{x}_l)$ pair, instead assuming that, on average, positive samples should have higher similarity scores than negative samples. To this end, we extend the InfoNCE loss to multiple instances:

$$\mathcal{L}^{e|l} = -\frac{1}{B_l} \sum_{k=1}^{B_l} \log \frac{\frac{1}{|P_k|} \sum_{j \in P_k} \exp(s_{j,k}^{e2l}/\tau)}{\sum_{j=1}^{B_e} \exp(s_{j,k}^{e2l}/\tau)} \quad (10)$$

$$\mathcal{L}^{l|e} = -\frac{1}{B_e} \sum_{k=1}^{B_e} \log \frac{\frac{1}{|Q_k|} \sum_{j \in Q_k} \exp(s_{j,k}^{l2e}/\tau)}{\sum_{j=1}^{B_l} \exp(s_{j,k}^{l2e}/\tau)} \quad (11)$$

$$\text{with } |P_k| \leq N \text{ and } |Q_k| \leq M, \quad (12)$$

$$\mathcal{L}^{e,l} = \frac{1}{2}\left(\mathcal{L}^{e|l} + \mathcal{L}^{l|e}\right) \quad (13)$$

and where $P_k$ and $Q_k$ are sets of positive EEG and text segments respectively. $B_e$ and $B_l$ are the batch sizes for EEG and text respectively, for which we sample up to $N$ EEG and $M$ text segments for $\frac{B}{N}$ subjects. We normalize using $|Q_k|$ or $|P_k|$ to account for the varying number of crops across subjects.

### 2.1.2. EEG-ONLY SELF-SUPERVISED LEARNING

We compare the representations learned by EEG-language pretraining to those obtained via EEG-only pretraining using the same pretraining dataset and EEG encoder. We

emphasize that "EEG-only" refers to pretraining without text, while ELMs use text solely during pretraining to guide EEG representation learning. At test time, neither method uses clinical reports, ensuring alignment with standard EEG-based clinical practice. We provide further information on the following methods in Appendix B.5: Bootstrap-Your-Own-Latent (BYOL; (Grill et al., 2020)), Variance-Invariance-Covariance Regularization (VICReg; (Bardes et al., 2021)), Contrast with the World Representation (ContraWR; (Yang et al., 2021)), Relative Positioning (RP; (Banville et al., 2021)), Temporal Shuffling (TS; (Banville et al., 2021)), Contrastive Predictive Coding (CPC; (Banville et al., 2021)).

## 3. Experimental Setup

### 3.1. Datasets and evaluation tasks

• **TUEG.** The Temple University Hospital (TUH) EEG Corpus is the largest available corpus of hospital EEG data with varying montages, channel counts, and sampling frequencies (n=26846 (Obeid & Picone, 2016)). For most of the dataset, no labels are available beyond patient age and sex. However, many EEG sessions are associated with a natural-language clinical report.

• **TUAB.** The TUH Abnormal EEG corpus is a subset of TUEG which was manually labeled by clinicians indicating whether the EEG displays pathological abnormalities (Lopez et al., 2015). This enables the binary classification task of predicting the status of {normal, abnormal} on a recording-level. Following the literature, we use the provided evaluation set as the hold-out test set.

• **NMT.** We leverage the NMT Scalp EEG Dataset (Khan et al., 2022) in order to validate our results for recording-level abnormality classification out-of-distribution. The NMT dataset deviates considerably from TUEG. Data was recorded from a South Asian population in Pakistan, using a different EEG recording setup. Furthermore, the NMT participants are considerably younger, feature more males (66.6%), and their EEG recordings are labeled predominantly normal (83.8% in the training set, while the test set is balanced). We use the provided train/test split.

To further evaluate learned representation, we use tasks requiring classification of single, short 5-second EEG crops.

• **TUSZ.** The TUH Seizure Detection Corpus (Shah et al., 2018) is a subset of TUEG which has sections labeled to contain either seizure or background activity. We perform binary classification using 5-fold cross validation on the provided train and dev sets (n=6491), while testing on the eval set.

• **TUEV.** An Events Corpus (Obeid & Picone, 2016) containing annotated EEG with six classes, of which three are

clinical (spike and slow wave, generalized periodic epileptiform discharge, periodic lateralized epileptiform discharge) as well as eye movements, artifacts, and background activity. We only use the provided train set (5-fold CV) due to the test set not including the TUEG subject identifiers, which would have prevented the exclusion of these subjects from the pretraining data. For each 1-second event, we include two seconds of context before and after (Jiang et al., 2024).

### 3.2. Preprocessing

#### 3.2.1. TEXT PROCESSING

In order to categorize the textual content in the clinical reports, we employed regular expressions matching for commonly-occurring headings (an overview is provided in Appendix E.4). These enabled the segmentation of individual reports into their respective headings with associated text paragraphs, providing insight into which information in physician reports is encoded in the EEG. We cluster headings into four categories, while filtering out headings which are irrelevant such as information on the EEG system, technical issues, or general disclaimers. First, the *clinical history* cluster of headings contains demographic information in terms of patient age and sex, as well as a brief description of relevant current and/or past pathology. The *record description* cluster includes the physician's observations of the EEG traces, which describes both normal and abnormal features, often in terms of oscillatory brain activity. The *medication* cluster contains the patient's current medication information. Finally, the *interpretation* cluster summarizes a physician's thoughts, often including the impression of whether the EEG is normal or pathologically abnormal, as well as a clinical correlation. To investigate whether EEG-language models can learn richer representations by being exposed to a larger variety of text, we also train models by sampling text from these four aforementioned clusters.

Due to the heterogeneity of the clinical reports, we further test the utility of summarizing the pathological status indicated by the clinical report using a large language model (LLM). Due to the sensitive nature of the clinical reports, we use the Llama-3 8B model (Meta, 2024) locally and instruct for the production of a single-sentence summary of a report, which should include whether the EEG was deemed abnormal and for which reasons (Appendix E.4).

**Language encoding.** Given a sampled section from a clinical report or the LLM-generated summary, we encode this text by relying on the embedding of the $[cls]$ token which aggregates the representations across all tokens. As such, given a clinical report $\mathbf{x}_l$, the transformation function $z_l$ corresponds to text segmentation or summarization yielding $\tilde{\mathbf{x}}_l$. Following tokenization, we embed into the $[cls]$ token using $f_l$. The resulting text embedding $\mathbf{h}_l$ may be used for multimodal pretraining.

#### 3.2.2. EEG PROCESSING

EEG data received minimal preprocessing, with our approach detailed in Appendix C.1. We describe the selection of our pretraining dataset in Appendix C.2, which avoids data leakage by excluding any data of subjects present in any of the evaluation data. The resulting sample sizes are shown in Table 1. To enable fair comparisons between methods, the optimal crop-length for recording-level pathology detection was determined out of $\{5,10,20,30,60\}$ seconds without data-leakage (Appendix B.2), yielding 20 and 60 second crops for EEG-only and EEG-language modeling respectively. For TUSZ and TUEV evaluations, we additionally pretrain models using 5-second crops and drop subjects which feature in either TUEV or TUSZ-test.

### 3.3. Pretraining setup

We set the temperature parameter $\tau$ to 0.3 using evaluation on a holdout set (Appendix B.4). For ELM-MIL, the number of sampled EEG and text crops per subject depend on the targeted downstream context. When using 60 second crops for subject-level prediction we set the number of sampled EEG crops $N = 32$ and text crops $M = 8$ paragraphs as this covers the available samples for a majority of subjects. For tasks using single 5-second crops, we increase $N$ to 120 and sample single sentences instead of paragraphs with $M = 24$ to accommodate the more finegrained nature of these tasks.

**Language encoder.** For $f_l$ we use a transformer model which was pretrained with medical data in a contrastive manner on PubMed search logs (MedCPT; (Jin et al., 2023)). See Appendix B.3 for a comparison of language models. $\text{ELM}_l$ adopts the language model's native hidden dimensionality (768), while for $\text{ELM}_{e,l}$ and ELM-MIL we project to a dimensionality of 256.

**EEG encoder.** For the EEG encoder $f_e$ we use a randomly initialized residual convolutional neural network, with an identical backbone architecture across all comparisons. We use nonlinear MLPs with a single-hidden layer for $g_e$ and $g_l$, as well as for the projector head in EEG-only self-supervised learning. More details are provided in Appendix B.2.

## 4. Experimental Results

### 4.1. Pretraining comparisons

Given the novelty of the studied domain, we extensively investigate the information represented in the learned embeddings resulting from EEG-language training as a function of both text selection and alignment strategy. To this end, we perform retrieval analyses as well as pathology detection using linear probes and zero-shot classification (Figure 2).

**Retrieval analyses.** Given a medical report describing

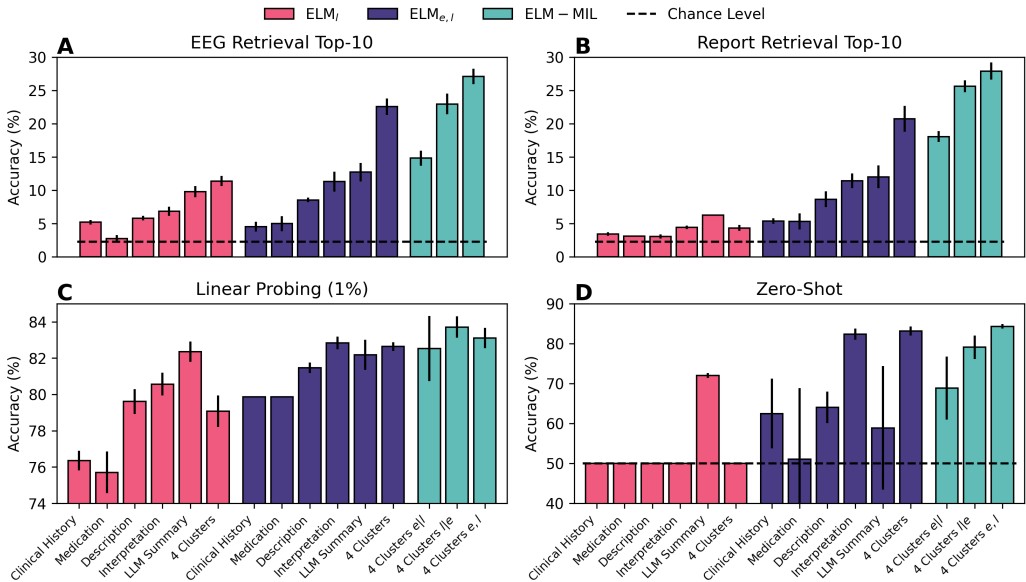

*Figure 2.* A,B) A set of EEG-Language models are evaluated on their retrieval ability using top-k accuracy out of 437 patients. C) Linear probing performance on TUAB with 1% of labels and D) zero-shot classification. Error bars indicate standard deviations over five model training runs.

*Table 1.* Dataset sample sizes. Crop lengths are indicated for ELM/EEG-only models.

| Data subset | EEG files | Reports | Crop Length |
|---|---|---|---|
| TUEG Pretrain | 15144 | 11785 | Variable |
| TUAB train | 2712 | Not used | 60/20s |
| TUAB test | 276 | Not used | 60/20s |
| Retrieval test | 437 | 437 | 60/- |
| NMT Train | 2216 | - | 60/20s |
| NMT Test | 183 | - | 60/20s |
| TUSZ Train+Dev | 6491 | - | 5/5s |
| TUSZ Test | 865 | - | 5/5s |
| TUEV Train | 359 | - | 5/5s |

the patient and their EEG recording, we probe the ability to recover the patient's EEG by rank-ordering candidate EEG based on embedding similarity, as well as vice versa. We average embeddings of EEG and text segments within-modality, yielding one EEG and report embedding per recording, which we use for rank-ordering based on cosine similarity.

Top-K retrieval accuracy, assessing if the correct EEG or report ranks within the top K (Figure 2A,B), shows that many models perform well above chance. This indicates successful generalization of multimodal EEG-language learning. Text sampling significantly impacts report retrieval; clusters lacking direct EEG descriptions (clinical history, medication) score lowest. Including descriptive information improves retrieval, but pathology-relevant contexts (inter-

pretation, LLM summary) are most effective, highlighting pathology as a key source of variation. Combining multiple text clusters further enhances results, suggesting unique information capture and the model's ability to integrate diverse patient data.

$ELM_{e,l}$ models tend to outperform $ELM_l$ models, particularly for report retrieval. This is likely due to omission of a text projection head in $ELM_l$, which may therefore lack the flexibility to appropriately separate the EEG reports in latent space. Due to the benefit of pretraining when sampling from the four text clusters, we pretrain our ELM-MIL models in this manner only. We observe that our MIL extension further improves retrieval performance, benefiting from sampling multiple positives jointly $(e, l)$ and performing bidirectional alignment. These results indicate for the additional flexibility of a MIL-based approach to aid in multimodal alignment, supporting the hypothesis that not all EEG and text pairs are equally informative.

**Clinical phenotyping.** We study the learned representations in their relevance to clinical pathology by training linear probes on TUAB (Figure 2C). Here we observe similar patterns for recording-level classification, with bidirectional ELM-MIL scoring best. Interestingly, good classification was possible with many of the models, causing us to investigate whether the strategy of sub-unit multimodal modeling provides inherent benefits. We provide this additional set of analyses in Appendix A.3, which indicates that our sub-unit alignment strategy promotes the encoding of between-subject information even in the absence of semantically

*Table 2.* Pathology detection via zero-shot (ZS) and linear probing at 1%, 10%, and 100% labeled data of the TUAB training set. The (second) best scores are printed (underlined) bold. Standard deviations over five model training runs are included. Supervised serves as a reference and refers to training end-to-end directly on labels.

| Method | Balanced Accuracy | | | | AUROC | | | |
|---|---|---|---|---|---|---|---|---|
| | ZS | 1% | 10% | 100% | ZS | 1% | 10% | 100% |
| Supervised | - | $71.36_{\pm1.10}$ | $81.06_{\pm0.30}$ | $84.13_{\pm0.29}$ | - | $79.87_{\pm1.30}$ | $89.23_{\pm0.51}$ | $91.83_{\pm0.32}$ |
| BYOL | - | $72.69_{\pm0.57}$ | $79.03_{\pm1.16}$ | $79.94_{\pm2.14}$ | - | $78.85_{\pm0.81}$ | $86.75_{\pm0.76}$ | $88.82_{\pm0.70}$ |
| VICReg | - | $71.76_{\pm0.81}$ | $79.6_{\pm1.07}$ | $82.46_{\pm0.96}$ | - | $78.7_{\pm1.11}$ | $86.04_{\pm0.80}$ | $88.78_{\pm1.04}$ |
| ContraWR | - | $73.30_{\pm1.44}$ | $80.72_{\pm1.69}$ | $82.44_{\pm1.22}$ | - | $80.30_{\pm1.91}$ | $86.67_{\pm1.32}$ | $88.44_{\pm1.20}$ |
| RP | - | $74.52_{\pm1.06}$ | $82.16_{\pm0.38}$ | $83.36_{\pm0.42}$ | - | $82.63_{\pm0.87}$ | $89.78_{\pm0.43}$ | $91.43_{\pm0.34}$ |
| TS | - | $74.99_{\pm0.86}$ | $82.16_{\pm0.64}$ | $84.10_{\pm0.66}$ | - | $82.51_{\pm0.91}$ | $89.58_{\pm0.55}$ | $91.50_{\pm0.32}$ |
| CPC | - | $73.20_{\pm0.79}$ | $78.44_{\pm1.00}$ | $79.95_{\pm1.49}$ | - | $81.48_{\pm1.02}$ | $86.44_{\pm1.07}$ | $87.92_{\pm1.14}$ |
| ELM-MIL $l\vert e$ | $68.86_{\pm7.89}$ | $82.53_{\pm1.80}$ | $\mathbf{86.38}_{\pm0.77}$ | $\mathbf{87.62}_{\pm0.43}$ | $75.23_{\pm9.28}$ | $89.88_{\pm1.47}$ | $92.92_{\pm0.54}$ | $93.52_{\pm0.34}$ |
| ELM-MIL $e\vert l$ | $\underline{79.10}_{\pm2.93}$ | $\mathbf{83.71}_{\pm0.59}$ | $\underline{84.37}_{\pm0.97}$ | $85.65_{\pm0.97}$ | $\underline{87.26}_{\pm3.19}$ | $\mathbf{92.37}_{\pm0.43}$ | $\mathbf{93.25}_{\pm0.27}$ | $\underline{93.65}_{\pm0.16}$ |
| ELM-MIL $e,l$ | $\mathbf{84.31}_{\pm0.57}$ | $\underline{83.10}_{\pm0.56}$ | $84.21_{\pm0.82}$ | $\underline{87.11}_{\pm0.76}$ | $\mathbf{91.56}_{\pm1.31}$ | $91.54_{\pm0.44}$ | $\underline{93.14}_{\pm0.24}$ | $\mathbf{93.91}_{\pm0.17}$ |

*Table 3.* Linear probing for abnormality classification on the NMT dataset using 1%, 10%, and 100% labeled training data.

| Method | AUROC | | |
|---|---|---|---|
| | 1% | 10% | 100% |
| BYOL | $63.78_{\pm1.70}$ | $76.48_{\pm2.10}$ | $80.65_{\pm2.50}$ |
| ContraWR | $\underline{65.72}_{\pm1.01}$ | $72.47_{\pm0.95}$ | $75.42_{\pm1.01}$ |
| VICReg | $61.57_{\pm1.49}$ | $74.19_{\pm0.63}$ | $78.50_{\pm1.60}$ |
| TS | $64.90_{\pm0.70}$ | $\underline{81.36}_{\pm1.53}$ | $\underline{87.08}_{\pm1.02}$ |
| RP | $64.92_{\pm0.81}$ | $80.42_{\pm1.83}$ | $86.50_{\pm2.17}$ |
| CPC | $65.24_{\pm2.06}$ | $77.84_{\pm1.12}$ | $79.98_{\pm1.60}$ |
| ELM-MIL | $\mathbf{69.49}_{\pm2.26}$ | $\mathbf{81.42}_{\pm1.15}$ | $\mathbf{89.77}_{\pm0.21}$ |

relevant text. This allows $ELM_{e,l}$ to nearly match the best EEG-only pretraining strategy for pathology detection when reports are randomly shuffled.

Next, we investigate the unique ability of multimodal language modeling to leverage the language modality to perform 'zero-shot' classification (Figure 2D). Without any explicit labels for downstream training, EEG may be classified by computing its similarity in latent space to text prompts representing the candidate classes. As suggested by Radford et al. (2021), we create a prompt ensemble over 21 variations of the phrasing "The EEG is normal, abnormal" (Appendix D). Despite a small dataset, EEG-language models can reach high levels of zero-shot pathology detection if: 1) they include a text projector, 2) include at least the clinician interpretation, and 3) do bidirectional alignment. These models, especially ELM-MIL$_{e,l}$, score highly consistent across different weight initializations, suggesting sound alignment between modalities. Pretraining using exclusively one of the other text clusters yielded poor performance, which follows from these models not being exposed to the explicit phrasing indicating the EEG status as normal or abnormal per se. Their capability likely can be improved

by designing appropriate prompts. Having identified the requirements for stable alignment, we focus on the ELM-MIL$_{e,l}$ model for EEG-only baseline comparisons due to its consistent performance across tasks and seeds.

### 4.2. Baseline comparisons

#### 4.2.1. ABNORMALITY CLASSIFICATION

We compare our ELM-MIL models on the TUAB dataset to six EEG-only methods (Table 2). We pretrain using these methods on the same pretraining dataset as ELMs and, together with the supervised baseline, also an identical EEG encoder architecture. This enabled accurate inference on the effectiveness of the pretraining strategy per se. We find ELMs yield large improvements for pathology detection over EEG-only pretraining, with multimodal models being particularly effective at small sample sizes: at 1% of exposed labels, performance increases reach 8.7% balanced accuracy and 9.7% AUROC. To validate our results out-of-distribution, we evaluate learned representations on the NMT dataset without finetuning (Table 3). We find ELM-MIL$_{e,l}$ (henceforth 'ELM-MIL') to still perform well, especially with few labels, where they yield an improvement of at least 3.8% AUROC.

Next, we further contextualize our results in the literature. By probing ELM-MIL representations for 10 second crop-level abnormality classification on TUAB with 80% labels we are able to compare to previously reported evaluations (Yang et al., 2024; Jiang et al., 2024; Dimofte et al., 2025). However, we note that methods use significantly different pretraining data and architectures, which complicates interpretation. Nevertheless, ELM-MIL scores highest and improves over LaBraM-Huge +0.83% on average across scores (Table 4). This is despite LaBraM-Huge requiring finetuning, being pretrained on a large number of datasets (including TUEG), and featuring 369M parameters com-

*Table 4.* Crop-level performance on TUAB (80% labels) of supervised and self-supervised methods using different training datasets and EEG encoders. Model sizes refer to trainable parameters for the EEG encoders.

| Methods | Fine tuned | Model Size | B. Acc. | AUROC |
|---|---|---|---|---|
| SPaRCNet | Y | 0.79M | $78.96_{\pm 0.18}$ | $86.76_{\pm 0.12}$ |
| ContraWR | Y | 1.6M | $77.46_{\pm 0.41}$ | $84.56_{\pm 0.74}$ |
| CNN-Transformer | Y | 3.2M | $77.77_{\pm 0.22}$ | $84.61_{\pm 0.13}$ |
| FFCL | Y | 2.4M | $78.48_{\pm 0.38}$ | $85.69_{\pm 0.51}$ |
| ST-Transformer | Y | 3.5M | $79.66_{\pm 0.23}$ | $87.07_{\pm 0.19}$ |
| BIOT | Y | 3.2M | $79.59_{\pm 0.57}$ | $88.15_{\pm 0.43}$ |
| CEReBrO | Y | 3.58M | $79.40_{\pm 0.19}$ | $87.49_{\pm 0.33}$ |
| CEReBrO | Y | 40.0M | $81.29_{\pm 0.15}$ | $88.67_{\pm 0.06}$ |
| CEReBrO | Y | 85.2M | $81.67_{\pm 0.23}$ | $89.16_{\pm 0.38}$ |
| LaBraM-Base | Y | 5.8M | $81.40_{\pm 0.19}$ | $90.22_{\pm 0.09}$ |
| LaBraM-Large | Y | 46M | $82.26_{\pm 0.15}$ | $91.27_{\pm 0.05}$ |
| LaBraM-Huge | Y | 369M | $82.58_{\pm 0.11}$ | $\mathbf{91.62}_{\pm 0.16}$ |
| ELM-MIL | N | 0.93M | $\mathbf{84.42}_{\pm 0.21}$ | $\underline{91.44}_{\pm 0.11}$ |

pared to 0.9M of our EEG encoder. To further compare the quality of learned representations, we apply our linear probing pipeline to the LaBraM-Base model (Appendix 7-9). We find gains for our ELM of up to 5.7–13.4% depending on the dataset, highlighting the improved semantic content in ELM representations. This stark discrepancy is likely due to the clinical specificity of ELMs, enabling much smaller models that do not need to be finetuned. As a result, our methodology is also significantly cheaper and faster to train (50 epochs in under 12 hours with 24GB of memory).

### 4.2.2. SEIZURE AND EVENT CLASSIFICATION

We additionally investigated classification performance based on single, five second EEG crops. Although recording-level classification was the focus of our work, we find ELM to score better than the investigated EEG-only methods also on both seizure detection (TUSZ) and 6-class event detection (TUEV; Table 6). We provide per-class performance in Appendix A.2.

### 4.2.3. FURTHER ABLATION ANALYSES

Whereas for InfoNCE the temperature parameter sets the relative focus across negative samples (Wang & Liu, 2021), for MIL-InfoNCE it does so too for positive samples. We therefore test the sensitivity of our methods to the parameter (Figure 3). We find that MIL-InfoNCE is more robust to changes of $\tau$ for pathology detection, while retrieval performance can be further improved by lowering $\tau$. This may be explained as retrieval being subject-based rather than class-based (see Appendix 13). Moreover, performance increases from $\tau < 1$ indicates the utility of this additional hyperparameter of InfoNCE, which is absent in NCE.

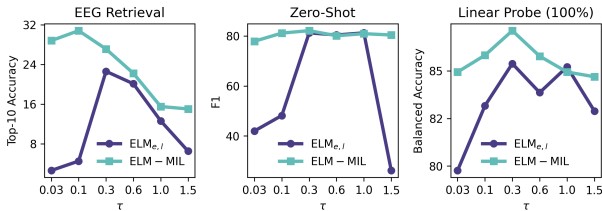

*Figure 3.* Model comparisons across EEG retrieval and pathology detection under different values of the temperature parameter $\tau$.

We perform additional ablations to investigate crucial aspects of the ELM-MIL $e, l$ model. First, we find that additional positive EEG and text samples improve downstream performance (Table 5). We additionally ablate the aggregation method for positive samples and find MIL-InfoNCE to outperform considered alternatives. We compare to aligning only the most similar positive sample (denoted Max+InfoNCE), using attention to create a weighted average across positive samples based on similarity values (Attn+InfoNCE; (Ilse et al., 2018)), as well as taking the sum instead of mean across log-probabilities (Sum+InfoNCE). The latter does not account for the varying amount of text and EEG crops across subjects.

### 4.2.4. ALIGNMENT VISUALIZATIONS

For model interpretability, we visualize temporal multimodal alignment in example hold-out recordings (Figure 4). We compute cosine similarity between all EEG crop embeddings in a recording and an embedded snippet from its paired clinical report, then plot crops with the highest and lowest similarity. We observe distinct periods of stronger/weaker alignment, where strong alignment corresponds to relevant clinical events. This indicates our ELMs achieve temporally selective alignment of clinical EEG events without explicit temporal event information or labels, consistently across independent pretraining runs. Additional examples, including failure cases, are in Appendix E.3.

## 5. Discussion

This paper presents a first application of multimodal pretraining combining natural language and functional brain data in a medical context. The proposed methodology achieves significantly improved clinical representations compared to EEG-only SSL, while being inexpensive to pretrain. This improvement stems from our novel sub-unit alignment approach in combination with MIL-InfoNCE to address inherent data misalignment challenges. Notably, these multimodal models enabled zero-shot pathology detection and label-efficient linear probing with improvements up to 9.7%. We additionally show generalization of ELMs via external validation and clinical event detection tasks. Our pathology-

*Table 5.* Ablation studies (Means over five training runs). Ret: EEG Retrieval (Top-10 accuracy), LP: Linear Probe (Balanced accuracy at 100%), ZS: Zero-shot classification (F1).

**(a) Aggregation**

| Method | Ret. | LP | ZS |
|---|---|---|---|
| Max+InfoNCE | 3.9 | 77.5 | 43.2 |
| Attn+InfoNCE | 8.3 | 84.9 | 17.5 |
| Sum+InfoNCE | 24.7 | 86.0 | 78.8 |
| MIL-InfoNCE | **27.1** | **87.1** | **82.1** |

**(b) Positive EEG Samples**

| N | Ret. | LP | ZS |
|---|---|---|---|
| 2 | 19.8 | 85.9 | 78.8 |
| 4 | 21.8 | 85.9 | 79.2 |
| 8 | 25.3 | 86.5 | 80.0 |
| 16 | 26.5 | 86.9 | 78.5 |
| 32 | **27.1** | **87.1** | **82.1** |

**(c) Positive Text Samples**

| M | Ret. | LP | ZS |
|---|---|---|---|
| 2 | **28.1** | 85.8 | 80.4 |
| 4 | 27.1 | 86.0 | 80.4 |
| 8 | 27.1 | **87.1** | **82.1** |

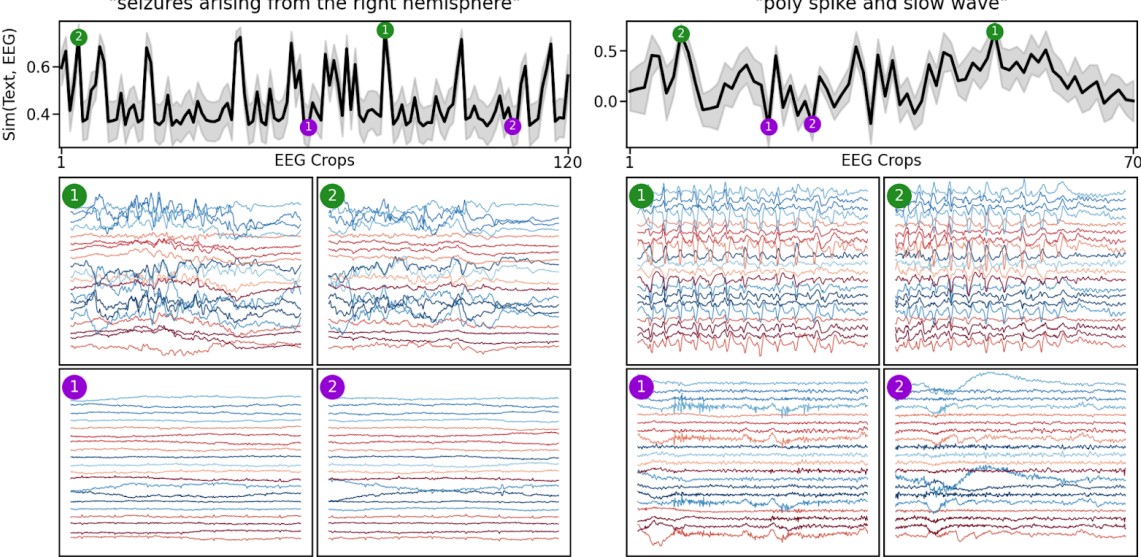

*Figure 4.* Left, top: The cosine similarity between EEG embeddings of 5-second crops and the embedding of the paired text "seizures arising from the right hemisphere" shows distinct periods of stronger and weaker alignment. Shading indicates the standard deviation across 5 random weight initializations. Green and purple dots mark the two crops with the highest and lowest similarity respectively. Left, bottom: EEG traces of the crops indicated by the coloured dots. Electrodes are colored based on their position (left: red, right: blue) and clearly reveal right-lateral seizure activity. Right: As left-sided plot, but for a different recording with the text "poly spike and slow wake".

*Table 6.* Linear probing for seizure and event classification on the TUSZ and TUEV datasets respectively.

| AUROC | TUSZ | | | TUEV |
|---|---|---|---|---|
| Method | 1% | 10% | 100% | 100% |
| Supervised | 84.00±0.90 | 87.33±0.93 | 90.38±0.39 | 86.28±0.78 |
| BYOL | 77.45±3.14 | 86.60±0.79 | 88.89±1.02 | 82.40±1.60 |
| ContraWR | 68.86±3.13 | 82.75±1.66 | 85.68±0.74 | 84.11±1.64 |
| VICReg | 69.61±1.82 | 82.01±0.93 | 86.17±0.98 | 83.26±1.83 |
| TS | 78.60±1.90 | 87.63±0.58 | 89.77±0.72 | 84.86±1.32 |
| RP | 64.41±2.50 | 76.72±1.37 | 79.52±1.26 | 78.95±1.59 |
| CPC | 72.01±1.00 | 81.77±2.24 | 85.91±2.00 | 81.94±1.75 |
| ELM-MIL | **78.98**±5.18 | **88.98**±0.86 | **91.51**±0.33 | **87.69**±1.01 |

sensitive multimodal alignment is a critical step toward automated report generation (e.g. Biswal et al. 2020), ensuring EEG-text representations capture clinical information for future documentation tasks.

Some considerations of this study deserve mention. The current limitation of publicly available paired EEG-report datasets presents a challenge for scaling pretraining data. Future work could address this through the generation of synthetic text captions based on clinical metadata. Furthermore, due to computational constraints, analyses on scaling model sizes are yet to be performed. However, we find highly favourable performance of ELMs compared to significantly larger EEG foundation models pretrained on a large number of datasets. We provide code and pretrained models at https://github.com/SamGijsen/ELM.

## Acknowledgements

This research was funded by the Deutsche Forschungsgemeinschaft (DFG) through FOR 5187 (project number 442075332). Additional support was provided by the DFG through the following projects: CRC 1404 (project number 414984028), TRR 265 (project number 402170461), and

RU 5363 (project number 459422098).

The data used in this study was provided by the Neural Engineering Data Consortium at Temple University. For further details about this data, please access the following URL: https://isip.piconepress.com/projects/tuh_eeg/html/.

The authors declare no competing interests.

## Impact Statement

Performance across both recording-level classification and event detection tasks suggests that our model learns clinically relevant features at multiple temporal scales. With further progress, this capability may support various future clinical applications, from rapid screening of prolonged recordings to real-time event detection. Whereas the present study focuses on establishing an initial application to explore viability, future work may benefit from focusing on improving the interpretability of these representations through techniques such as channel-specific attribution. The multimodal nature of our approach, by aligning EEG with clinical reports in a pathology-sensitive manner, not only enhances detection but also lays an important foundation for automated report generation. Specifically, such generation may greatly benefit from an aligned latent space which contains clinical information. This could facilitate clinical documentation by translating EEG signals into structured summaries. These can constitute highly valuable future efforts given the time-intensive nature of manual reporting. Certain clinical limitations also deserve further attention, such as a careful study of how the frequency of specific pathology and clinical events in reports impacts model performance. Finally, biases present in language models may impact multimodal pretraining, which should be investigated in future work.

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

# A. Additional Results

## A.1. Linear probing comparisons with LaBraM

We apply our linear probing pipeline to the LaBraM-Base model and find significant performance drops, highlighting its requirement for downstream finetuning and the improved clinical representations of ELMs. We perform these analyses for NMT (Table 7), TUAB (Table 8), and TUEV (Table 9), while omitting TUSZ as this data was included in LaBraM pretraining. Standard deviations indicate variability across five repetitions of 10-fold cross-validation. Whereas for ELM-MIL we use a model trained with different weight initialization for each repetition, for LaBraM only one pretrained model is made available which we use for every repetition.

Table 7. Linear probing for abnormality classification comparing ELM-MIL and LaBraM on the NMT dataset using 1%, 10%, and 100% labeled training data.

| | Balanced Accuracy | | | AUROC | | |
|---|---|---|---|---|---|---|
| Method | 1% | 10% | 100% | 1% | 10% | 100% |
| LaBraM | $59.40_{\pm1.07}$ | $\mathbf{69.12}_{\pm0.73}$ | $67.52_{\pm0.11}$ | $66.13_{\pm1.77}$ | $78.03_{\pm0.59}$ | $82.02_{\pm0.21}$ |
| ELM-MIL | $\mathbf{60.60}_{\pm0.54}$ | $68.57_{\pm0.90}$ | $\mathbf{81.00}_{\pm1.18}$ | $\mathbf{69.49}_{\pm2.26}$ | $\mathbf{81.42}_{\pm1.15}$ | $\mathbf{89.77}_{\pm0.21}$ |

Table 8. Linear probing for abnormality classification comparing ELM-MIL and LaBraM on the TUAB dataset using Zero-Shot, 1%, 10%, and 100% labeled training data.

| | Balanced Accuracy | | | | AUROC | | | |
|---|---|---|---|---|---|---|---|---|
| Method | ZS | 1% | 10% | 100% | ZS | 1% | 10% | 100% |
| LaBraM | - | $72.71_{\pm1.46}$ | $80.33_{\pm0.23}$ | $82.18_{\pm0.11}$ | - | $80.18_{\pm0.81}$ | $87.98_{\pm0.35}$ | $89.61_{\pm0.01}$ |
| ELM-MIL | $84.31_{\pm0.57}$ | $\mathbf{83.10}_{\pm0.56}$ | $\mathbf{84.21}_{\pm0.82}$ | $\mathbf{87.11}_{\pm0.76}$ | $91.56_{\pm1.31}$ | $\mathbf{91.54}_{\pm0.44}$ | $\mathbf{91.91}_{\pm0.17}$ | $\mathbf{93.14}_{\pm0.24}$ |

Table 9. Linear probing for abnormality classification comparing ELM-MIL and LaBraM on the TUEV dataset using 80% labeled training data.

| | BACC | AUROC |
|---|---|---|
| Method | 80% | 80% |
| LaBraM | $43.08_{\pm1.65}$ | $83.22_{\pm1.09}$ |
| ELM-MIL | $\mathbf{48.83}_{\pm2.80}$ | $\mathbf{87.69}_{\pm1.01}$ |

## A.2. TUEV: Per-Class analysis

We additionally investigated per-class performance as TUEV includes distinctly different event categories (Figure 5). We observe that ELM-MIL scores well across the three clinical events (SPSW, GPED, PLED) with over 3.5% better average scores. However, the model underperformed on artifact and eye movement detection, which may indicate models may lose sensitivity to events not described in the text. Interestingly, a portion of reports include sections on such technical problems, but these were segmented out for the current study. Follow-up research is needed to further investigate the effects of including such text.

## A.3. Language-independent effects of sub-unit alignment.

**Language-independent effects of sub-unit alignment.** Given the broad outperforming of ELMs compared to EEG-only models, especially for $\text{ELM}_{e,l}$, we further investigate whether the general setup of multimodal pretraining provides inherent benefits. EEG recordings are split into multiple crops, which in turn are all aligned to the same clinical report during pretraining. It follows that EEG crops of a single recording are indirectly aligned to one another to some extent (Figure 1C). We investigated this hypothesis by shuffling reports between patients prior to pretraining. We find that while embeddings of single EEG crops of an untrained encoder are only minimally more similar within-subject than between-subject (ratio of

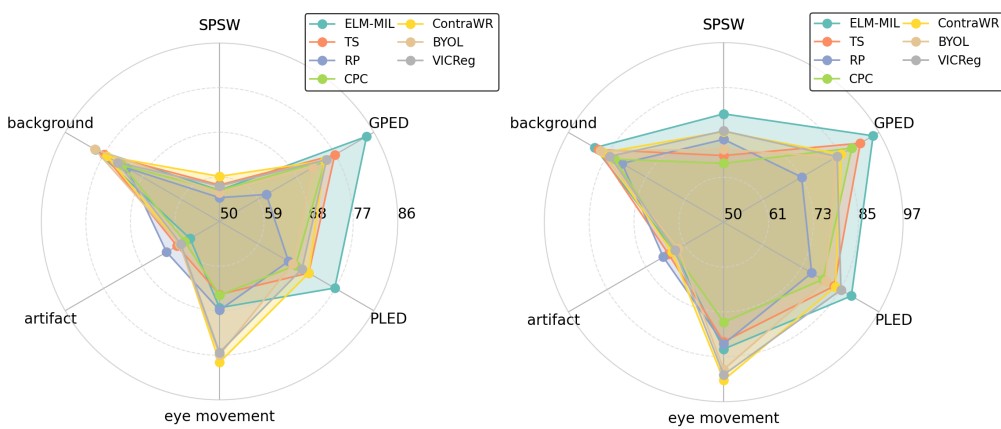

Figure 5. Per-class scores for TUEV show that ELM-MIL outperforms for the clinical events. SPSW: Spike and sharp wave, GPED: generalized periodic epileptiform discharges, PLED: periodic lateralized epileptiform discharges.

∼1.1x), this effect is much more pronounced after pretraining $ELM_{e,l}$ on correctly paired reports (∼6.3x), and even more so after pretraining on shuffled reports (∼15.7x; figure 6). Linear probing reveals that training $ELM_{e,l}$ on shuffled reports clearly boosts pathology detection over using an untrained encoder and manages to almost match EEG-only pretraining without the need for augmentations (mean accuracies of 73.70%, 81.04%, 83.69%). On the contrary, the ratios for $ELM_l$ are close to 1 after training using paired and shuffled reports, with the latter resulting in decreased pathology detection accuracy.

Conceptually, while shuffling reports destroys the semantic relevance of reports, it still provides a unique subject-specific reference to which the EEG embeddings are aligned to. Pretraining then reduces to promoting invariance to within-subject information, as all EEG crops of a patient are aligned to the same report. However, while for $ELM_l$ these reports occupy arbitrary positions in the latent language space due to the absence of the text projector, $ELM_{e,l}$ exhibits additional dynamics. Namely, for a given EEG crop (or text paragraph) in a batch belonging to subject $i$ (that is, $id = i$), nearly all negative contrastive samples will belong to a different patient ($P(id = i) \ll P(id \neq i)$). The negative contrast therefore largely amounts to minimizing similarity between patients. This can be viewed as encoding between-subject information and these results imply that training with this objective is a useful pretext task for EEG timeseries. Naturally, this will depend on the downstream tasks, but both retrieval and pathology detection require between-subject information. The advantage of retrieval and linear probing of $ELM_{e,l}$ may thus be, at least in part, due to the inherent utility of our extension of multimodal language modeling to timeseries by using sub-unit alignment, independent of language. Still, pathology detection with only few annotations is considerably better using paired reports, indicating the importance of relevant clinical language for label-efficiency.

### A.4. Post-hoc investigation of data leakage

To maximize the amount of data available in this data-scarce setting, the TUAB training set was included during pretraining. We investigate whether this gave a disproportionate advantage to linear probes trained on ELM representations by repeating the "1% labels" context using unseen subjects as follows: Given only the TUAB test set, we train linear probes using 10-fold cross validation (times five random seeds), each time splitting 10-20-70% of the test set into train/validation/test. This gives the same labeled sample size as 1% of the TUAB training set without relying on samples seen during pretraining. As seen in Table 10, results are highly similar, strongly suggesting that the advantage of ELMs is not due to the inclusion of the TUAB training set in the pretraining set.

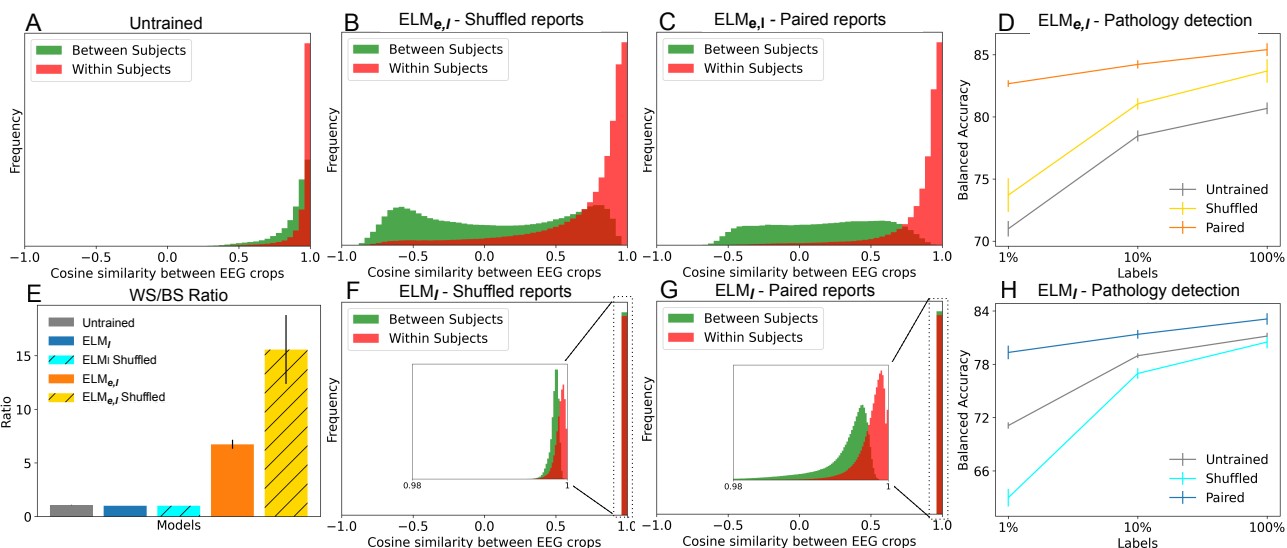

*Figure 6.* **A-C, E-G)** We investigate the distributions of cosine similarity values of EEG crop embeddings between- and within subjects (denoted BS and WS respectively). We plot these for an untrained encoder (one example run), as well as EEG encoders of ELMs trained with paired or shuffled reports. We find that $ELM_{e,l}$ produces dissimilar between-subject EEG embeddings, while $ELM_l$ does not. **E)** shows the ratio between WS and BS similarity values across five runs (with standard deviations). **D,H)** The downstream performance via linear probing is shown on the right, with error bars representing standard deviations across five training runs.

*Table 10.* Effect of overlap in subjects used for pretraining and linear probing. Higher standard deviations result from a smaller test set.

| Method | Overlap | Balanced Accuracy |
|---|---|---|
| TS | Yes | $74.99_{\pm 0.86}$ |
| TS | No | $74.56_{\pm 1.12}$ |
| $ELM_{e,l}$ 4 Clusters | Yes | $82.64_{\pm 0.24}$ |
| $ELM_{e,l}$ 4 Clusters | No | $82.28_{\pm 0.64}$ |

## B. Training Details

In this section, we provide further detailed information of the model training. Unless stated otherwise, ablation and hyperparameter analyses were performed on a data subset consisting of 5000 and 500 EEG recordings divided into a training and test set respectively. To prevent data leakage, this data had no overlap with the patients used for evaluation of the main results.

### B.1. Optimization

All models are pretrained using the LARS optimizer (You et al., 2017) with a cosine decay learning rate schedule over 50 epochs, with a warm-up of 4 epochs. The base learning rate is set to 0.3 for EEG-only, 0.01 for ELMs, and 0.06 for ELM-MIL, scaled with the batch size (BaseLR × BatchSize/256; (Grill et al., 2020)). When training on 5-second crops, we lower the learning rate for EEG-only to 0.1 and ELM-MIL to 0.02 to avoid instability. We use a weight-decay parameter of $1 \times 10^{-4}$. Models were trained on either an Nvidia Geforce GTX 3090 or Tesla V100 GPU and require less than 24GB of memory. Training took approximately 9 hours for EEG-language modeling or 18 hours for EEG-only modeling due to data augmentations. We used CUDA v11.3 and PyTorch v1.12.1.

### B.2. EEG Encoder

We use a CNN architecture with a residual stream as the EEG encoder for all analyses (Figure 7). The model uses parallel convolutions, involving reflection padding and 1D-convolutions with kernel sizes $\{4, 8, 16\}$ with 32 filters each. These

outputs are concatenated, resulting in a 96 dimensional representation and 747K trainable parameters. We compare input lengths of EEG crops varying from 5 to 60 seconds. This presents a trade-off where longer crops result in a greater information content per crop, while reducing the total sample size. As EEG-only pretraining relies on data augmentations, this introduces an additional influence of crop length. Specifically, longer crop lengths likely make the pretraining task easier, as augmentations introduce relatively lesser distortion due to the greater information content. We therefore compare performance of different crop lengths for both EEG-language and EEG-only pretraining. As the EEG encoder progressively downsamples the signal, we adjust the pooling layers to the input length. These adjustments are shown in Table B.2. For EEG-language pretraining we evaluate zero-shot pathology detection, while for EEG-only pretraining we are required to compare the performance of a linear probe. Results are shown in Figure 8. Due to computational resources, we only compare crop lengths for BYOL and ELM$_l$ as representations of EEG-only and EEG-language modeling. We observe that for EEG-only pretraining an intermediate crop-length of 20 seconds performs best, which matches the findings by Mohsenvand et al. (2020). Meanwhile, zero-shot pathology detection is found to be relatively insensitive to crop lengths of at least 10 seconds, with 60 second crops scoring highest, while the shortest crop length showed unstable learning. For TUSZ and TUEV evaluations, this was solved by lowering the learning rate.

For the EEG projector, we use a linear layer with an output dimension of 512 followed by batch normalization, exponential linear units, and a final linear layer with output size 256.

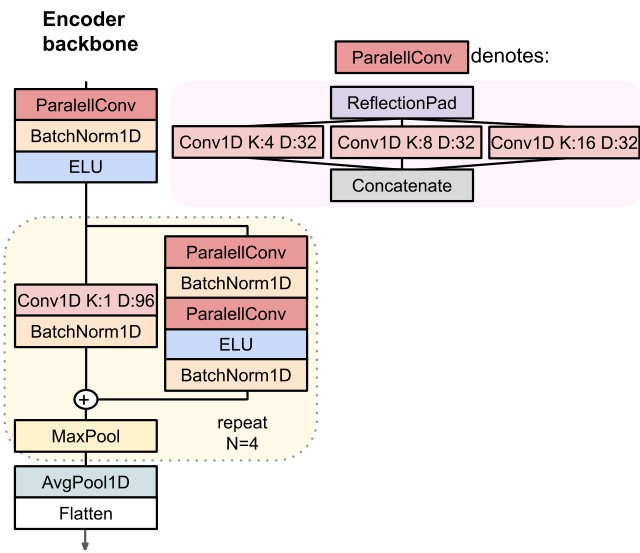

*Figure 7.* An identical EEG encoder architecture is used across all analyses. The size of the max pool operation depends on the input length. These are detailed in table B.2. K: Kernel size, D: Output dimensionality.

*Table 11.* Multiple input lengths for the cropped EEG timeseries were compared, which included adjustments to the pooling layer.

| | Model Setups | | Batch Size | |
|---|---|---|---|---|
| Input Dim. | Max Pool Size | Intermediate Dim. | EEG+Text | EEG |
| 500 | [2,2,2,2] | [166, 55, 18, 6] | 2048 | 2048 |
| 1000 | [3,3,3,3] | [333, 111, 37, 12] | 2048 | 2048 |
| 2000 | [3,3,3,3] | [666, 222, 74, 24] | 2048 | 1024 |
| 3000 | [4,4,4,4] | [750, 187, 46, 11] | 1024 | 800 |
| 6000 | [4,4,4,4] | [1500, 375, 93, 23] | 800 | 400 |

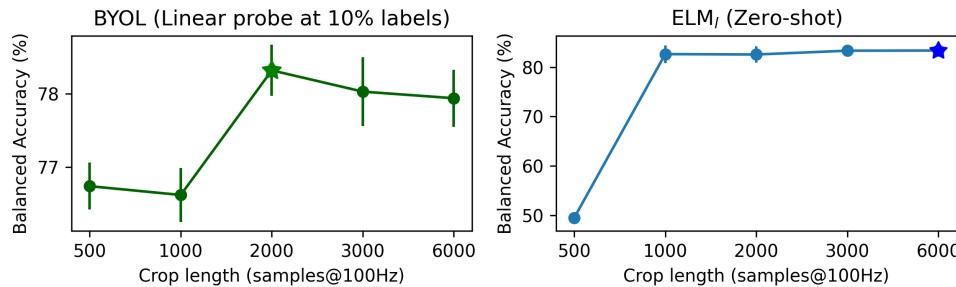

*Figure 8.* Comparison of pathology detection based on EEG input crop length, ranging from 5 to 60 seconds, via averaged balanced accuracy scores. Error bars indicate the standard deviation across five random seeds.

## B.3. Language Encoder

We compare three pretrained language models in their ability to perform zero-shot pathology detection following EEG-language pretraining (Table 12). We find that MedCPT performs best (Jin et al., 2023), which is trained using contrastive learning with 255 million user click logs from PubMed.

For the text projector of $ELM_{e,l}$, we use a linear layer with output size 1024 followed by batch normalization, rectified linear units, and a final linear layer with output size 256 and batch normalization.

*Table 12.* Zero-shot classification comparison between language models for $ELM_{e,l}$.

| Language Model | Balanced Accuracy | AUROC |
|---|---|---|
| BiomedBERT (Gu et al., 2021) | $78.61_{\pm 2.90}$ | $85.78_{\pm 2.58}$ |
| Bio-ClinicalBERT (Alsentzer et al., 2019) | $80.86_{\pm 1.19}$ | $87.33_{\pm 0.68}$ |
| MedCPT (Jin et al., 2023) | $\mathbf{82.58}_{\pm 0.25}$ | $\mathbf{88.37}_{\pm 0.39}$ |

## B.4. Temperature parameter

For $ELM_{e,l}$, the softmax operation used in the loss computation includes a temperature hyperparameter $\tau$. We compare zero-shot pathology detection for multiple values. We observe poor performance for low temperature values, but stable zero-shot classification for higher parameter values. We set $\tau = 0.3$ for all further analyses.

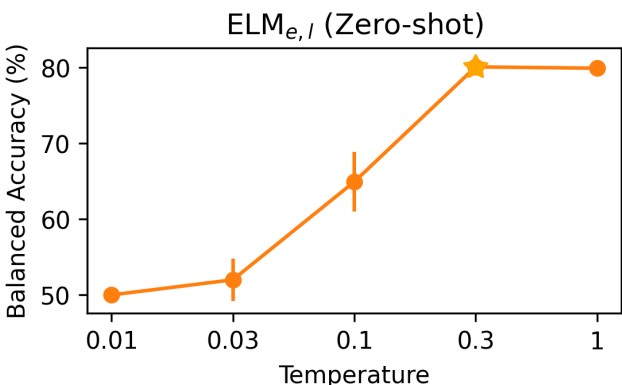

*Figure 9.* Comparison of temperature values for $ELM_{e,l}$ on zero-shot pathology detection. Error bars indicate the standard deviation across three random seeds.

## B.5. EEG-Only Pretraining

We implement the following methods for EEG-only SSL:

**Bootstrap-Your-Own-Latent.** BYOL relies on two encoder models: an online and a target network (Grill et al., 2020). During pretraining, the online network is trained to predict the target model's output. Meanwhile, the weights of the target network are updated using a moving average of the weights of the online network, which has been empirically shown to prevent collapse of the latent space. For alignment, $\ell_2$ normalization is applied to the EEG embeddings $\{\mathbf{h}'_e, \mathbf{h}''_e\}$ and the mean square distance is minimized. We adopt the recommended parameter value for the exponential moving average (Grill et al., 2020). The projection head is a 2-layer non-linear MLP with a hidden dimension of width 256 and an output dimension of 32.

**Variance-Invariance-Covariance Regularization.** VICReg allows for the use of a single encoder model and prevents collapse by applying two explicit regularization terms to each of the embedding batches $\{\mathbf{h}'_e, \mathbf{h}''_e\}$ (Bardes et al., 2021). The 'variance' term maintains the standard deviation (computed batch-wise) of every embedding dimension above a threshold, thereby avoiding a trivial solution. In addition, latent collapse is avoided through the 'covariance' term which decorrelates pairs of embedding dimensions. The method minimizes the mean square distance between $\{\mathbf{h}'_e, \mathbf{h}''_e\}$. Hyperparameters are set to their recommended values (Bardes et al., 2021). The projection head is a 2-layer non-linear MLP with a hidden dimension of width 256 and an output dimension of 256.

**Contrast with the World Representation.** ContraWR was proposed to improve augmentation-based SSL for EEG (Yang et al., 2021). The method, which is contrastive in nature, maximizes similarity between $\{\mathbf{h}'_e, \mathbf{h}''_e\}$ while preventing collapse by minimizing similarity with 'negative' samples. ContraWR forms a negative representation by aggregating across all negative batch elements, aiming to compensate for the low signal-to-noise of EEG data by creating a more reliable negative contrast. It relies on a triplet loss based on Info-NCE (Gutmann & Hyvärinen, 2010). We also here set the hyperparameters to the values recommended by the authors (Yang et al., 2021). The projection head is a 2-layer non-linear MLP with a hidden dimension of width 256 and an output dimension of 32.

**Relative Positioning**. Pairs of EEG crops are sampled and assigned binary labels based on their temporal proximity (Banville et al., 2021). Crops close in time are labeled positive, while those far apart are labeled negative. We use the same EEG encoder as for all other methods to create representations and use the suggested contrastive module to compute the element-wise absolute difference between representations. A logistic regression model then predicts the label. The method is trained using binary logistic loss. For all methods by (Banville et al., 2021), we use the hyperparameters reported to work best on TUAB, including between-subject sampling of EEG crops.

**Temporal Shuffling**. An extension of Relative Positioning by sampling triplets of EEG crops. The task is to determine whether the crops are in temporal order or shuffled (Banville et al., 2021). The contrastive module concatenates absolute differences between representations. As with Relative Positioning, a logistic regression model is used for prediction, and the method is trained end-to-end using binary logistic loss.

**Contrastive Predictive Coding**. This method uses an autoregressive encoder to summarize a sequence of EEG crops into a context vector (Banville et al., 2021). The task is to predict which future crop actually follows the context, among negative samples. A bilinear model is used for prediction at each future step. The method is trained end-to-end using the InfoNCE loss.

### B.5.1. DATA AUGMENTATIONS

For EEG-only pretraining, we adapt the data augmentations proposed by Mohsenvand et al. (2020), which were found to perform well on the TUAB dataset. For a given EEG crop, we apply the same augmentation to each channel. Parameters are sampled independently for each EEG crop and uniformly from the ranges displayed in table 13. Augmentations are visualized for a single EEG channel in figure 10.

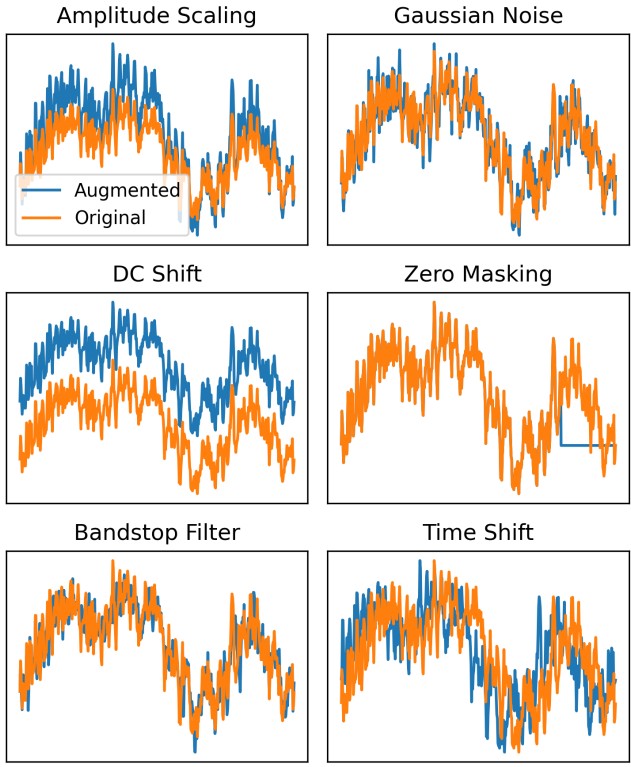

*Figure 10.* Data augmentations visualized for a single channel of EEG data.

*Table 13.* Data augmentation parameter ranges; adapted from Mohsenvand et al. (2020).

| **Data Augmentation** | **Min** | **Max** |
|---|---|---|
| Amplitude Scale | 0.5 | 1.5 |
| Time Shift in samples | -60 | 60 |
| DC shift in microvolts | -10 | 10 |
| Zero-Masking in samples | 0 | 200 |
| Additive Gaussian Noise (sigma) | 0 | 0.2 |
| Band-Stop Filter (5Hz width, Hz) | 2.8 | 47 |

## C. Details on EEG Data

### C.1. EEG preprocessing

From the EEG dataset, recordings longer than 2.5 hours were omitted to filter out a small subset of very long, potentially overnight recordings. For training efficiency, only the first 45 minutes of a recording were used. Any recording files shorter than 70 seconds were also omitted.

EEG data received minimal preprocessing (using MNE (Gramfort et al., 2013)). First, the initial 10 seconds were removed to reduce the impact of set-up artefacts. Afterwards, a bandpass filter of 0.1-49 Hz was applied and all recordings were resampled to 100 Hz. To reduce the impact of signal artefacts, all EEG signals had their amplitude clipped to $\pm$ 800 $\mu$V. As a large majority of recordings used an average-reference (AR) or linked-ear reference (LE), we only used these recordings and standardized them via transformation to the 20-channel Temporal Central Parasagittal (TCP) montage.

## C.2. Subsampling

TUEG contains considerably more abnormal than normal EEGs. As vision-language models have been shown to be sensitive to imbalanced classes (Wang et al., 2024), we subsample the data to create approximately equal class representation. We rely on the LLM summaries of reports, which were more consistent in their phrasing regarding the normal or abnormal status. This allowed for a more reliable classification using regular expressions. All reports for which no clear classification was made were omitted. 5015 reports in the potential training set were classified as normal, which were associated with 7526 EEG recordings. For our 'pretrain' data subset, we subsampled the abnormal EEGs to match the amount of normal EEG recordings. This resulted in 7526 abnormal EEG recordings, with 6770 reports. Although only a minor subset of these preliminary classifications was manually verified, it is important to note that this process was solely to alleviate severe class imbalance and was not used for further analysis.

For EEG-language modeling, the pretrain subset was effectively smaller, as a report had to be omitted from pretraining when it did not contain at least one relevant heading. Out of the 15144 total EEG files, this resulted in pretrain sample sizes of: 14836 (clinical history cluster), 14320 (medication cluster), 14800 (description cluster), 14794 (interpretation cluster), and 14946 (four clusters).

To test for retrieval performance, we supplemented the TUAB test set with data from the TUH EEG Epilepsy Corpus (Veloso et al., 2017) in an attempt to create a larger, roughly balanced evaluation set of those with and without pathology. For this, we only selected the first recording of a subject so that no multiple files from the same subject were present. Additionally, we only included reports which had at least one heading from each text cluster to allow for a fair comparison.

# D. Classification

To study the predictive capability of learned representations after pretraining, we train linear probes and perform zero-shot classification.

### Linear probe

For linear evaluation, we train logistic linear regression models using 10-fold cross validation for each pretrained model using sklearn (Pedregosa et al., 2011). We perform grid-search over 45 logarithmically-spaced values for L2 regularization between $10^{-6}$ and $10^5$ via a validation set.

### Supervised Learning

For the supervised learning baseline, we use the identical EEG encoder backbone as used for all other analyses and use 60 second crops. We add an MLP (hidden dimensionality of 256) with dropout $p = 0.5$ and output dimensionality equal to the amount of classes. The ADAM learning rate is set to 0.001 and we use the validation set to select weight decay out of $[0.1, 0.01, 0.0001]$. We use a batch size of 256 and train using the cross entropy loss. When using 100% labels, we first train on the training set for up to 50 epochs (with early stopping after 5 epochs without improvement) and select the epoch which resulted in the best validation loss. Subsequently, we continue training on the train and validation sets together until the loss has decreased below the best validation loss.

### Zero-shot classification

For zero-shot pathology detection, we perform an ensemble over 21 binary prompts, listed in Table 14. Prompt ensembling was shown to improve performance (Radford et al., 2021), but we employ it here also as the limited data is likely to lead to less stable representations, which may lead to sensitivity to phrasing. To inspect whether results are sensitive to changes to the prompt set, we perform a post-hoc analysis using the held-out test set that iteratively leaves one prompt out of the ensemble (Figure 11). We observe that results are consistent across such reduced prompt sets, except for the $ELM_l$ model trained on the clinical history or interpretation clusters, although neither model reaches competitive performance. This set was only initially verified on the training set to enable model- and parameter-comparisons using zero-shot performance. Tuning is likely to enable further performance improvements, although the flexibility of the zero-shot approach may introduce severe risk of overfitting on the TUEG dataset.

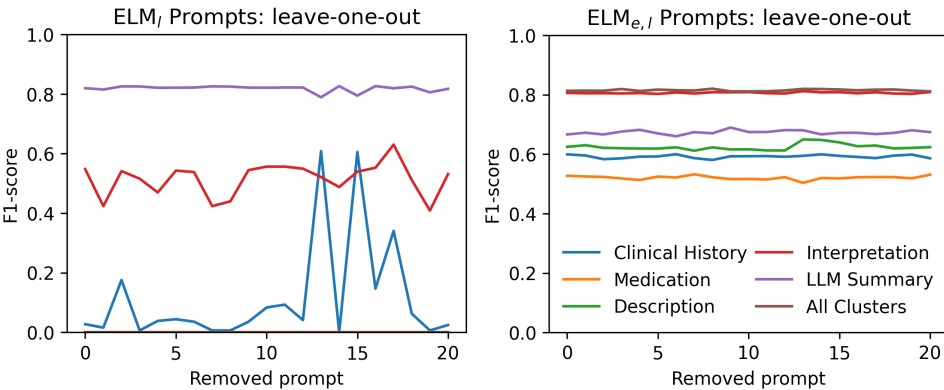

*Figure 11.* Analysis of the sensitivity to prompts in the ensemble used for zero-shot classification. We plot the average F1-score across five random seeds. Note that for $ELM_l$, multiple models have a consistent F1-score of 0 and are therefore not individually visible.

*Table 14.* Prompt ensemble used for zero-shot classification.

| Normal EEG Prompts | Abnormal EEG Prompts |
|---|---|
| Normal EEG. | Abnormal EEG. |
| No pathology present. | Pathology present. |
| No abnormalities. | Abnormalities observed. |
| Normal routine EEG. | Markedly abnormal EEG. |
| Normal awake record. | Abnormal awake record. |
| Normal EEG record. | Abnormal EEG record. |
| This EEG is normal. | This EEG is abnormal. |
| This is a normal EEG. | This is an abnormal EEG. |
| This EEG is within normal limits | This EEG is mildly abnormal. |
| Normal awake EEG. | Abnormal awake EEG. |
| Normal asleep EEG. | Abnormal asleep EEG. |
| Normal awake and asleep EEG. | Abnormal awake and asleep EEG. |
| Normal EEG in wakefulness and drowsiness. | Abnormal EEG in wakefulness and drowsiness. |
| No pathology. | Abnormal EEG due to: |
| EEG shows no pathology. | Abnormal EEG for a subject of this age due to: |
| No abnormalities. | Abnormalities in the EEG. |
| No abnormalities observed. | Abnormalities observed. |
| EEG shows no abnormalities. | EEG shows abnormalities. |
| No clinical events detected. | Clinical events detected. |
| No indications of pathology observed. | Indications of pathology observed. |
| The EEG is normal. | The EEG is pathologically abnormal. |

# E. Additional visualisations

## E.1. EEG embeddings of pathology

We provide t-SNE (complexity=40, (Van der Maaten & Hinton, 2008)) visualisations of the averaged EEG embeddings per subject after pretraining. These are post-hoc plots for which we use models trained on the entire pretraining subset and display embeddings of hold-out TUAB patients. $ELM_{e,l}$ and ELM-MIL show the clearest visual separation between abnormal and normal EEGs, which is in line with the linear probing results.

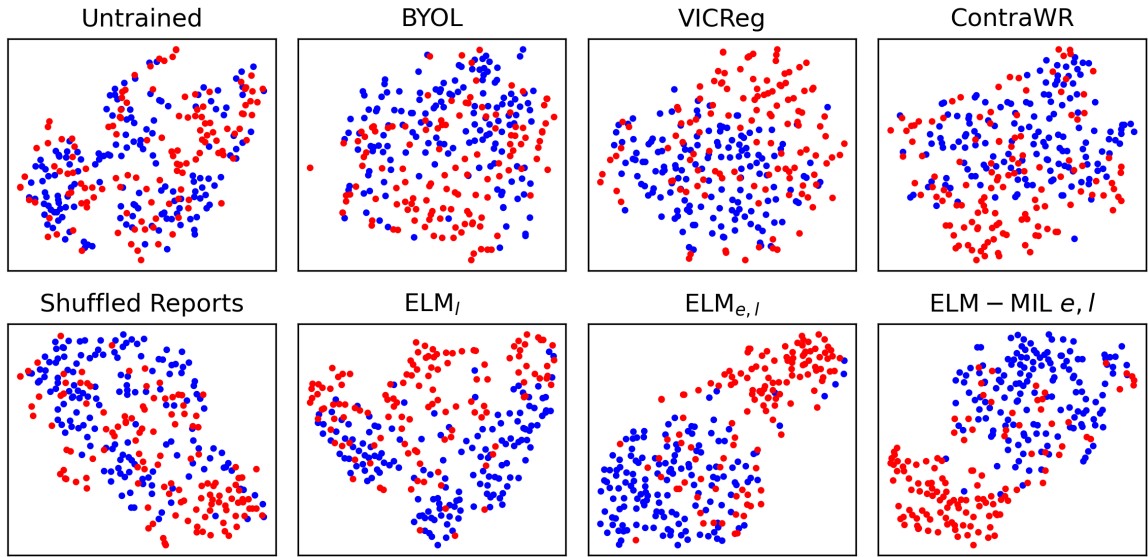

*Figure 12.* Example EEG embeddings averaged within-subject of pretrained models on the TUAB hold-out data (red: abnormal, blue: normal). The data is projected using t-SNE. The 'untrained' and 'shuffled reports' plots feature the same setup as the $ELM_{e,l}$ model, with the latter being trained on reports randomly shuffled between subjects.

### E.2. Within-subject EEG embeddings

We provide additional visualizations of t-SNE projections of EEG crops (Figure 13). Specifically, we compare $ELM_{e,l}$ using InfoNCE and ELM-MIL using MIL-InfoNCE across three temperature parameters $\tau = [0.1, 0.3, 1.0]$. To do so, we randomly sample three normal (blue shades) and three abnormal (red shades) subjects. We observe that whereas both methods exhibit diminished subject clustering at a higher temperature ($\tau = 1.0$), at low temperatures ($\tau = 0.1$) this only occurs for InfoNCE. Meanwhile, subject clustering gets more pronounced for MIL-InfoNCE. This may explain the observation that retrieval performance increases by reducing $\tau$ for MIL-InfoNCE, which as a task requires subject rather than class separation per se.

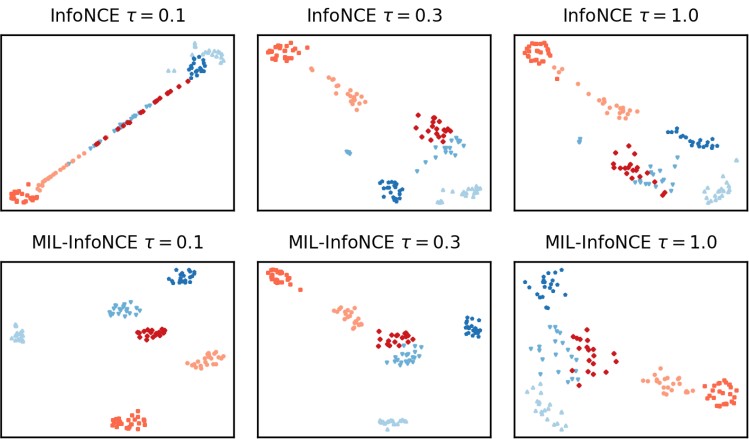

*Figure 13.* A comparison of subject clustering using t-SNE projections of embeddings of EEG crops. Red (blue) shades indicate three randomly sampled abnormal (normal) subjects.

## E.3. Alignment visualizations

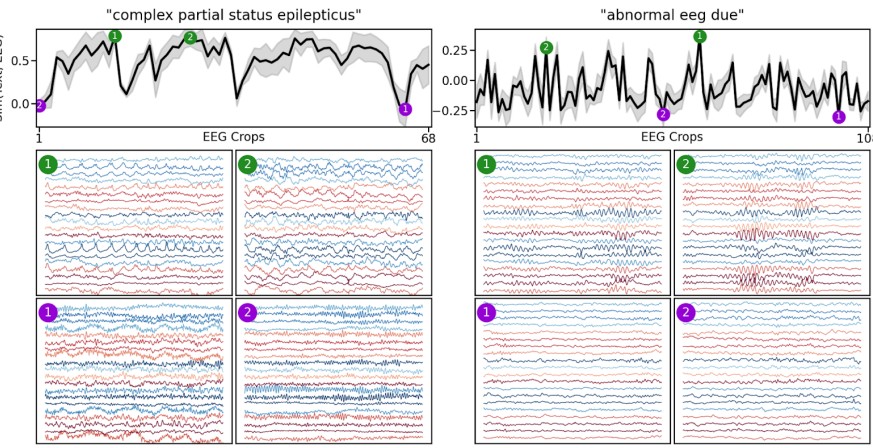

Figure 14. Additional alignment visualizations as in Figure 4.

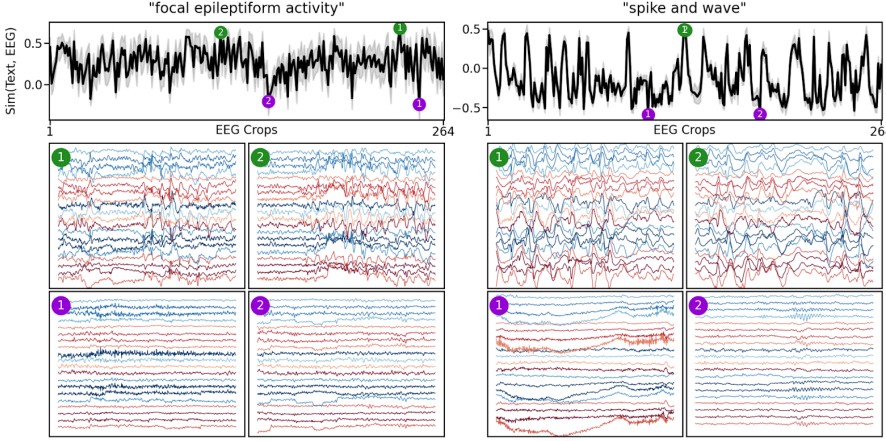

Figure 15. Additional alignment visualizations as in Figure 4.

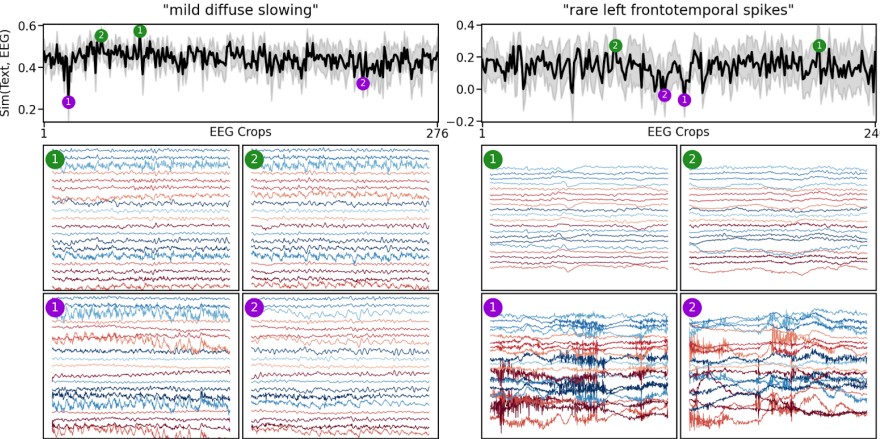

Figure 16. Additional alignment visualizations as in Figure 4, but for two observations which were not well aligned. Both feature a narrow range of similarity values, indicating a lack of temporal localization. Whereas mild diffuse slowing may have not been pronounced enough, left frontotemporal spikes are very rarely described in the dataset.

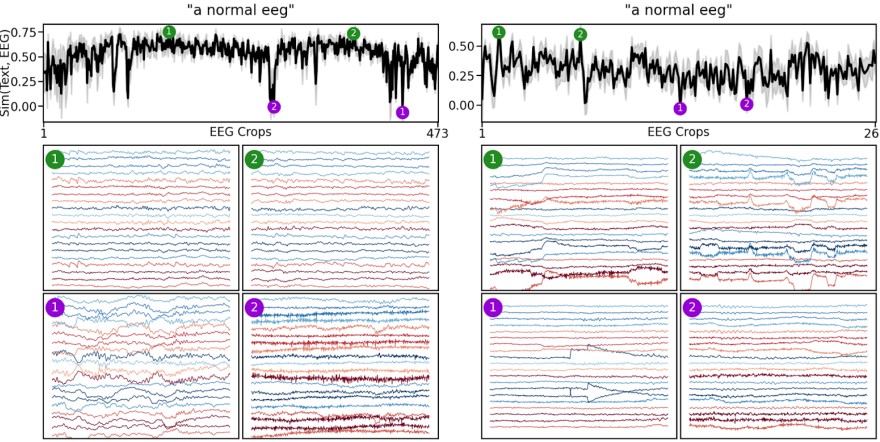

*Figure 17.* Additional alignment visualizations as in Figure 4, but for two recordings which were classified as normal by the physician.

## E.4. Report and content segmentation

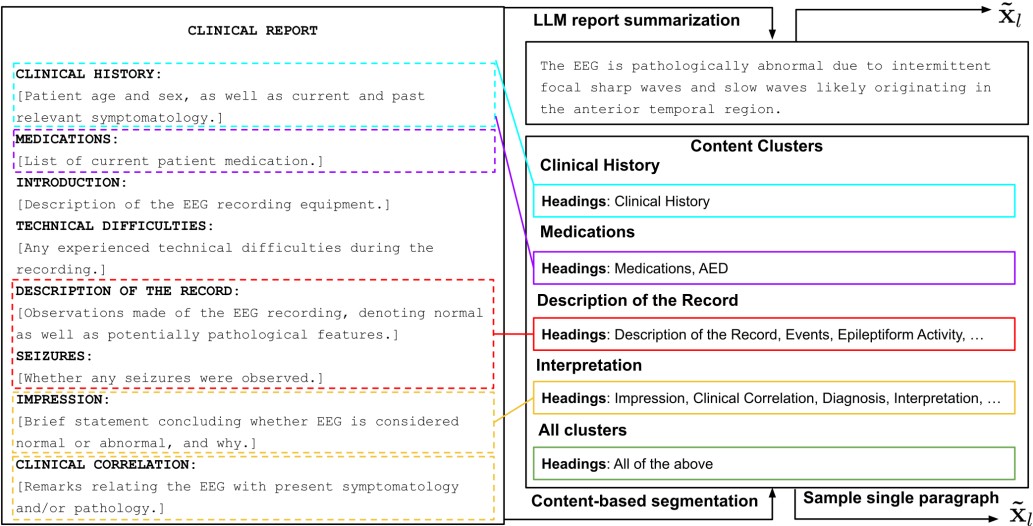

*Figure 18.* An example set of headings which may make up a clinical report. Paragraphs are extracted from the reports into content-based clusters or an LLM-generated summary.

