# OpenReview forum: "EEG-Language Pretraining for Highly Label-Efficient Clinical Phenotyping"
_ICML.cc/2025/Conference — ICML 2025 poster_

### Official Review · Reviewer_Qj1r · 2025-03-12

**Overall Recommendation:** 4

**Summary:**

This paper introduces EEG-Language Models (ELMs), a multimodal framework that integrates EEG signals with clinical text reports for various downstream tasks, including retrieval, abnormality classification, and event classification, across multiple datasets. The method employs time-series cropping, text segmentation, and a multiple-instance learning (MIL) variant of contrastive learning to address the alignment challenges between EEG signals and textual descriptions. Experimental results show that ELMs outperform EEG-only pretraining methods. Notably, the model exhibits zero-shot capability, further highlighting its adaptability across diverse downstream tasks. These findings underscore the potential of multimodal pretraining in medical applications, enabling richer and more effective representations for classification and retrieval tasks.

**Claims And Evidence:**

The experiments and analyses presented in the paper are sufficient to substantiate the authors' contributions.

**Essential References Not Discussed:**

No

**Experimental Designs Or Analyses:**

The experimental design is reasonable, with a diverse set of datasets and downstream tasks. The proposed method is compared against both supervised and self-supervised models, which strengthens the validity of the results. However, the paper only includes large-scale EEG models like LaBraM as a baseline in table 4, while the SSL baselines for other tables are relatively outdated. It is recommended to include more recent models and large-scale EEG models to provide a more comprehensive comparison.

**Methods And Evaluation Criteria:**

The proposed methodology aligns well with the research problem and is well-motivated. The evaluation framework is comprehensive, with multiple benchmark datasets and appropriate performance metrics. The chosen baselines allow for a fair comparison, effectively demonstrating the advantages of the proposed approach.

**Other Comments Or Suggestions:**

In Tables 2 and 6, the abbreviation "SV" is not explicitly defined, which may lead to ambiguity. It would be helpful to provide the full term for clarity.

**Other Strengths And Weaknesses:**

Strengths:
•	Novelty: The paper introduces an innovative approach by integrating EEG and language modeling with multiple-instance learning.
•	Methodological soundness: The proposed methodology is well-grounded and effectively addresses EEG-text alignment challenges.
•	Comprehensive experiments: The study evaluates the model across multiple tasks, demonstrating strong performance.
Weakness:
•	Baseline comparison: The paper only includes large-scale EEG models like LaBraM as a baseline in table 4, while the SSL baselines for other tables are relatively outdated. It is recommended to include more recent models and large-scale EEG models to provide a more comprehensive comparison.

**Questions For Authors:**

No major questions. The paper is well-structured, and the methodology is clearly explained.

**Relation To Broader Scientific Literature:**

The paper presents a novel approach by integrating EEG with language modeling and multiple-instance learning. It builds upon prior works such as CLIP and M-FLAG while incorporating multiple-instance learning to better handle EEG-text alignment. The experimental results convincingly demonstrate the effectiveness of this framework, highlighting its potential for advancing EEG applications in the medical domain. The study makes a meaningful contribution by extending multimodal learning techniques to EEG-based medical analysis.

**Theoretical Claims:**

The paper is primarily empirical and does not involve rigorous theoretical proofs.

---

> ### Author Rebuttal · Authors · 2025-03-31
>
> Dear Reviewer,
>
> Thank you for your thorough and supportive review of our manuscript. We are grateful for your positive assessment of our work’s novelty, methodological soundness, and comprehensive evaluation, as well as your recommendation to accept. We have used your constructive suggestions to refine our paper.
>
> **Baseline comparisons**
> We appreciate your comment regarding the baseline comparisons, particularly the limited inclusion of large-scale EEG models like LaBraM beyond Table 4. While we initially hesitated due to challenges in isolating the effects of data, encoder, architecture, parameter counts, or pretraining strategy, we recognize the value of broader comparisons. Following your suggestion, we have extended our evaluation to include LaBraM across additional datasets (TUAB subject level, NMT, and TUEV, while omitting TUSZ as they pretrain on this dataset) using the same evaluation strategy we used for all other methods. We adopted LaBraM’s preprocessing recommendations (resampling, bandpass filtering, notch-filtering, avoiding bipolar montages, and crop lengths of 10s for TUAB/NMT, 5s for TUEV) and obtained EEG embeddings. The updated results, visible in Tables S1-3 (available at this [link](https://docs.google.com/document/d/e/2PACX-1vQygdcAED1qMhVgFv5jU9TsclAyRxp-XKFiGwxK2pkxLSdrKAgyGVuAEYBVPnmQZeJDfIBVMLTbzTwG/pub)), show that ELM-MIL outperforms LaBraM across clinical contexts, with accuracy gains of up to 5.7–13.4% depending on the dataset. Recent literature such as LaBraM has predominantly focused on pretraining on many datasets (about 20 in their case) with large transformers to yield general representations. However, these representations are nonspecific and not ideal for downstream prediction tasks without fully finetuning the encoder, which is problematic when limited downstream data is available as in clinical contexts. In contrast, using medical text during pretraining helps ELMs learn relevant representations, to which we ascribe their strong performance.
>
> **Abbreviation**
> Additionally, thank you for noting the undefined abbreviation “SV” in Tables 2 and 6. We apologize for the oversight—“SV” refers to “Supervised”—and now provide the full term to ensure clarity.
>
> **Alignment visualisations**
> Finally, we would like to kindly note the addition of alignment visualisations, which may be found in Figures S1-5 at the same [link](https://docs.google.com/document/d/e/2PACX-1vQygdcAED1qMhVgFv5jU9TsclAyRxp-XKFiGwxK2pkxLSdrKAgyGVuAEYBVPnmQZeJDfIBVMLTbzTwG/pub). These highlight the ability for our method to localize pathology, while also indicating shortcomings such as rarely mentioned features. We hope the reviewer finds these informative.  We add Figure S1 to the main text along with a short paragraph noting the successes and shortcomings, while adding Figure S2 through S5 to the appendix.
>
> We hope these revisions address your suggestions effectively. Thank you once again for your insightful feedback.

---

### Official Review · Reviewer_eiWw · 2025-03-15

**Overall Recommendation:** 3

**Summary:**

This paper introduces an approach for pretraining multimodal EEG-language models (ELMs) to improve pathology detection. The authors propose combining EEG data with clinical reports using a sub-unit alignment strategy, which involves cropping EEG time series and segmenting medical reports to create multiple non-overlapping samples. They further extend this approach with multiple instance learning (MIL) to address misalignment between EEG and text segments. The proposed model significantly improves pathology detection performance, especially in scenarios with limited labels. These results are particularly applicable to clinical settings, where datasets are typically much smaller than those in many common deep learning applications.

## update after rebuttal

Thanks for the authors' rebuttals. The comments have addressed most of my concerns. I would keep my score.

**Claims And Evidence:**

The paper makes a clear claims with convincing evidence.

**Essential References Not Discussed:**

In the paper, the authors have thoroughly discussed various works related to EEG-Language Models (ELMs), including the latest advancements in self-supervised learning, multimodal modeling, and EEG-based pathology detection. However, the related work section could be further expanded by including research on alignment strategies in multimodal learning.
Alignment strategies are crucial in multimodal learning. While the paper mentions methods such as CLIP and M-FLAG, there have been recent advancements in multimodal alignment. These methods could provide new insights for improving EEG-text alignment.

**Experimental Designs Or Analyses:**

The experimental designs and analyses are sound and well-executed. The authors conduct extensive experiments to validate their approach, including:
Retrieval Analysis: The authors evaluate the ability of ELMs to retrieve matching EEG recordings from clinical reports and vice versa, using top-K accuracy as the metric.
Pathology Detection: The authors compare ELM-MIL to EEG-only models on the TUAB dataset for binary classification of normal vs. abnormal EEG recordings. They also evaluate performance on the NMT dataset to assess generalization.
Zero-shot Classification: The authors demonstrate zero-shot classification performance using a prompt ensemble, showing that ELMs can leverage language embeddings for pathology detection without explicit downstream training.
Ablation Studies: The authors conduct ablation studies to investigate the impact of different components, such as the aggregation method for positive samples and the number of positive EEG/text samples.
The experimental designs are comprehensive and address various aspects of the proposed approach. The results provide clear evidence of the effectiveness of ELM-MIL in improving pathology detection.

**Methods And Evaluation Criteria:**

The proposed methods and evaluation criteria make sense.

**Other Comments Or Suggestions:**

N/A

**Other Strengths And Weaknesses:**

Strengths:
Innovative Approach: The paper presents a novel application of multimodal pretraining in the medical domain, combining EEG data with clinical reports in a meaningful way.
Significant Improvements: The proposed ELM-MIL model demonstrates substantial improvements in pathology detection, especially in scenarios with limited labeled data.
Zero-shot Classification: The ability to perform zero-shot classification using language embeddings is a unique and powerful feature of the proposed approach.
Weaknesses:
Data Limitations: The availability of paired EEG-report datasets is limited, which may restrict the scalability of pretraining data. Future work could explore synthetic data generation techniques.
The related literature is insufficient. The related work section could be expanded by including research on alignment strategies in multimodal learning. Alignment strategies are crucial in multimodal learning. While the paper mentions methods such as CLIP and M-FLAG, there have been recent advancements in multimodal alignment. These methods could provide new insights for improving EEG-text alignment.
The paper lacks logical structure. The methods section should focus solely on describing the methodology. For instance, the statement “We set N=32 and M=8 as this covers all samples for a majority of subjects.” refers to specific experimental parameters and should be moved to the experiments section.

**Questions For Authors:**

In practical applications, the dataset size may be much larger. In this case, would the model's training time and memory requirements increase significantly?
The multimodal model proposed in the paper performs excellently in pathology detection, but model interpretability is crucial for clinical applications. Has the author considered improving the model's interpretability through specific techniques, such as feature importance analysis or attention mechanism visualization?

**Relation To Broader Scientific Literature:**

In the "Related Work" section, the authors thoroughly discuss the connections between their research and the broader scientific literature, highlighting key contributions in relation to previous studies. This section is divided into four subsections: Self-supervised learning with EEG data, Using EEG for pathology detection, Medical multimodal language modeling, and Multiple instance learning.
The proposed EEG-Language Models (ELMs) introduce multimodal pretraining by aligning EEG with text, significantly improving pathology detection performance, particularly in scenarios with limited labeled data. This represents a major advancement compared to previous self-supervised learning (SSL) approaches that rely solely on EEG data.

**Theoretical Claims:**

The paper does not include a clearly defined theoretical proof section. However, in the methodology section, it provides a detailed description of how concepts such as multimodal alignment, time-series cropping, text segmentation, and multiple instance learning are applied to the pretraining of EEG and language models. The theoretical foundation of these methods primarily stems from existing research in contrastive learning and multimodal representation learning. For example, the paper mentions the InfoNCE loss function and MIL-InfoNCE loss function, both of which are based on the theoretical framework of contrastive learning and are used to learn aligned representations between EEG and text. These theoretical frameworks have been extensively studied and validated in other domains, making the proposed methods theoretically sound.
However, the paper could further discuss the specific application and adaptability of these theoretical frameworks in EEG and language pretraining. For instance, while the multimodal alignment strategy and MIL extension are theoretically designed to mitigate inconsistencies between EEG and text segments, such inconsistencies may still persist in practical applications, especially when clinical reports contain a large amount of information unrelated to downstream clinical tasks.

---

> ### Author Rebuttal · Authors · 2025-03-31
>
> Dear Reviewer,
>
> Thank you for your detailed and constructive review of our manuscript. We greatly appreciate your recognition of our approach’s innovation, performance improvements, and clinical relevance, as well as your thoughtful suggestions and questions, which have helped us strengthen our work.
>
> **Alignment and challenges**
> We appreciate your suggestion to further discuss our EEG-language pretraining including its potential challenges with alignment. While our MIL extension aims to mitigate inconsistencies, challenges may persist when reports contain information unrelated to clinical tasks. However, our results (e.g., robustness to additional text sections in Figure 2) suggest this is less impactful for common clinical concepts. We are pleased to provide new model intepretability figures which visualize alignment (Figures S1-5 available at this [link](https://docs.google.com/document/d/e/2PACX-1vQygdcAED1qMhVgFv5jU9TsclAyRxp-XKFiGwxK2pkxLSdrKAgyGVuAEYBVPnmQZeJDfIBVMLTbzTwG/pub)). While we show that our methodology is able to localize pathology despite a lack of explicit temporal information (Figures S1-S3), alignment for rare or subtle features remains challenging (Figure S4). These examples highlight both successes and shortcomings, setting the stage for future refinements. We add Figure S1 to the main text along with a short paragraph noting the successes and shortcomings, while adding Figure S2 through S5 to the appendix.
>
> **Multimodal literature**
> Regarding literature on multimodal alignment, we are expanding the "Related Work" section to include further literature. In case the reviewer believes we are omitting important relevant work, we would be sincerely grateful for further suggestions.
>
> Related Work - Medical multimodal language modeling. [L100] (...) *Recent advances outside the medical domain include multi-task strategies, both during pretraining by integrating contrastive learning and self-supervised losses [1,2], as well as finetuning on multiple downstream tasks [3,4]. Further exploration involves moving compute from unimodal encoding to multimodal fusion [5].*
>
> [1] Tschannen, Michael, et al. "Siglip 2: Multilingual vision-language encoders with improved semantic understanding, localization, and dense features." arXiv preprint arXiv:2502.14786 (2025).
> [2] Tang, Zineng, et al. "TULIP: Towards Unified Language-Image Pretraining." arXiv preprint arXiv:2503.15485 (2025).
> [3] Liu, Haotian, et al. "Improved baselines with visual instruction tuning." Proceedings of the IEEE/CVF Conference on Computer Vision and Pattern Recognition. 2024.
> [4] Dai, Wenliang, et al. "InstructBLIP: Towards General-purpose Vision-Language Models with Instruction Tuning." arXiv, 2023, arxiv.org/abs/2305.06500.
> [5] Kim, Wonjae, Bokyung Son, and Ildoo Kim. "Vilt: Vision-and-language transformer without convolution or region supervision." International conference on machine learning. PMLR, 2021.
>
> **Logical structure**
> We appreciate the note about the logical structure and the placement of phrases such as “We set N=32 and M=8”. To improve the structure of the manuscript, we propose the following sections:
> 3. Methods (unchanged)
> 3.1 Pretraining (was: Experimental Setup)
> 3.1.1 EEG-language pretraining (unchanged)
> 3.1.2 EEG-only self-supervised learning (unchanged)
> 4. Experimental Setup (new)
> 4.1 Pretraining setup (new)
> 4.2 Datasets and evaluation tasks (was 3.2)
> 4.3 Preprocessing (was 3.3)
>
> and move the mentions of hyperparameters settings (such as N, M, temperature, model dimensionality) to '4.1 Pretraining Setup'. This separates the description of the methodology and experimental details into distinct sections. We thank the reviewer for their suggestion.
>
> **Training scalability**
> Regarding training scalability, our approach benefits from a small EEG encoder (0.9M parameters) compared to large-scale models like LaBraM (up to 369M), a frozen text model, and no finetuning requirement. This efficiency enables training on large batches with a single GPU (e.g., 9 hours on our dataset using a two-generation-old GPU) and thus scalability should remain very manageable.
>
> **Interpretability**
> For interpretability, we agree on its clinical importance and add the aforementioned alignment visualizations. This provides interpretability in the temporal domain and we hope the reviewer finds them valuable. While for the current manuscript we used an efficient CNN encoder to enable a clear focus on comparisons between pretraining strategies per se, we recognize that scaling to transformer architectures with attention visualizations can further enhance interpretability across the spatial domain. We leave these important encoder architecture explorations to follow-up work.
>
> We hope these comments address your feedback effectively. Thank you again for your insightful comments, which have significantly improved our manuscript.

---

### Official Review · Reviewer_hXr5 · 2025-03-18

**Overall Recommendation:** 1

**Summary:**

This paper presents a multi-modality model that integrates EEG recordings and clinical reports for neural event detection. The proposed method segments an EEG recording and its corresponding report into sequences of epochs and words, then constructs epoch-word pairs and an alignment matrix for representation learning. The model employs both pairwise contrastive learning and multi-instance contrastive learning to enhance feature representation. Finally, it is fine-tuned for various downstream tasks. Experiments on multiple datasets demonstrate that the proposed approach outperforms several contrastive learning and EEG-based baselines.

**Claims And Evidence:**

My main concern is the motivation behind aligning EEG recordings with clinical reports. EEG is a time-series signal that records a patient real-time physiological state, such as sleep stages, seizure events, and other neurological conditions. However, clinical reports typically consist of structured sections, such as an abstract and findings/details, which provide a summarized interpretation of the EEG recording. These reports often include patient information, the state during recording (e.g., awake, asleep, or under stimulation), and observations of seizure activity or specific waveforms, but they lack precise time indices, e.g., onset/offset of neural events.

Authors also point out this lack issue of time information of clinical reports but attempt to address it by using a neural network to learn representations of report segments and force-align them with temporal EEG epochs. However, this approach lacks clinical feasibility, as there is no inherent one-to-one correspondence between EEG epochs and report segments. Simply learning latent representations for alignment without considering the structured, non-temporal nature of clinical reports does not meet real-world clinical workflows.

**Essential References Not Discussed:**

CLARA: Clinical Report Auto-completion, WWW20.

**Experimental Designs Or Analyses:**

Please kindly refer to Methods And Evaluation Criteria.

A  meaningful task involving clinical reports would be report generation, as the authors also highlight in the Discussion and Impact Statement section. However, the current focus on detection and classification contradicts the intended use of clinical reports.

**Methods And Evaluation Criteria:**

Partially, the proposed method focuses on detection/classification tasks, but incorporating additional detailed clinical report, which require manual writing by doctor, significantly increases data costs. Moreover, such neural detection tasks can naturally be performed using EEG data alone, without the necessity of aligning with textual reports. The added complexity of learning from reports does not provide clear motivation and advantages for classification tasks.

**Other Comments Or Suggestions:**

Please kindly refer to the above comments

**Other Strengths And Weaknesses:**

Please kindly refer to the above comments

**Questions For Authors:**

Please kindly refer to the above comments

**Relation To Broader Scientific Literature:**

EEG-Text alignment is a promising research direction. However, constructing a benchmark dataset, designing an effective alignment method, and establishing a meaningful evaluation framework remain open challenges. Addressing these aspects would be highly beneficial for advancing clinical tasks and BCI applications.

**Theoretical Claims:**

NA

---

> ### Author Rebuttal · Authors · 2025-03-31
>
> Dear Reviewer,
>
> Thank you for your detailed and insightful review of our manuscript. We appreciate your feedback as it has helped us refine our presentation and clarify the motivations behind our work. We are encouraged by your recognition of EEG-text alignment as a promising research direction and would like to address your concerns.
>
> **EEG-Text Alignment**
> We acknowledge the point about the lack of explicit temporal correspondence between EEG epochs and clinical report segments, which indeed poses a challenge. However, our approach leverages the fact that concepts like “abnormal EEG” in reports are guaranteed to relate to multiple EEG crops from the same recording and, in expectation, more so than to unrelated (negative) crops. This principle mirrors successful contrastive learning in computer vision (e.g., noisy image-caption pairs) and video-text alignment (e.g., subtitles not always matching visuals), where pretraining remains effective despite imperfect correspondence. To illustrate this, we have added qualitative examples of our ELMs at this [[link]](https://docs.google.com/document/d/e/2PACX-1vQygdcAED1qMhVgFv5jU9TsclAyRxp-XKFiGwxK2pkxLSdrKAgyGVuAEYBVPnmQZeJDfIBVMLTbzTwG/pub) (Figures S1-5), showing similarity scores between text embeddings (e.g., “seizures arising from the right hemisphere”) and 5-second EEG crops for hold-out subjects. Visualizations of the highest- and lowest-similarity crops demonstrate that our method captures pathology-relevant alignment in the absence of explicit temporal information in the reports, reinforcing its practical utility. We add Figure S1 to the main text and Figure S2 through S5 to the appendix.
>
> **Downstream tasks and data costs**
> Regarding the motivation for downstream tasks, we agree that EEG-only methods can address detection/classification. However, our multimodal approach enhances unimodal EEG encoder initialization, yielding better representations for these tasks. As shown in our manuscript, our method outperforms EEG-only baselines (e.g., +8.7% balanced accuracy at 1% labels on TUAB) and even large-scale models like LaBraM (Table 4, as well as Tables S1-3 with new additional linear probing results at the same [link](https://docs.google.com/document/d/e/2PACX-1vQygdcAED1qMhVgFv5jU9TsclAyRxp-XKFiGwxK2pkxLSdrKAgyGVuAEYBVPnmQZeJDfIBVMLTbzTwG/pub)), trained on many more EEG datasets. The pretrained EEG encoders we are releasing can be used for downstream EEG-only clinical tasks without additional complexity and at reduced cost due to the low parameter count.
>
> This suggests that leveraging existing clinical reports—already available in large quantities in hospitals and with neglible costs to store compared to EEG data and especially compared to acquiring additional EEG data—offers a cost-effective way to improve performance without increasing data collection costs. Moreover, we like to clarify that our method repurposes existing clinical reports, and does not require new manual writing. These reports, a byproduct of standard clinical workflows, enhance model performance without additional expense.
>
> **Report generation**
> Finally, we appreciate your suggestion of report generation as a valuable task, and we fully intend our work to pave the way for such applications, as noted in the manuscript. We believe that learning and evaluating pathology-sensitive representations is a critical first step; if alignment fails to capture clinical relevance in latent representations, subsequent text generation would lack grounding. Our focus on detection/classification validates this foundation, aligning with the broader utility of clinical reports. To emphasize how our results establish a stepping stone for future tasks like report generation, we now note this in the discussion and expand the impact statement to contextualize this trajectory.
>
> **Discussion addition [L422]:**
> Our pathology-sensitive multimodal alignment is a critical step toward automated report generation (e.g. Biswal et al. 2020), ensuring EEG-text representations capture clinical information for future documentation tasks.
>
> **Impact statement extension [L457]:**
> The multimodal nature of our approach, by aligning EEG with clinical reports in a pathology-sensitive manner, not only enhances detection but also lays an important foundation for automated report generation. Specifically, such generation may greatly benefit from an aligned latent space which contains clinical information. This could facilitate clinical documentation by translating EEG signals into structured summaries. These can constitute highly valuable future efforts given the time-intensive nature of manual reporting.
>
> We hope these clarifications address your concerns and demonstrate the clinical and scientific value of our approach. Thank you again for your constructive feedback, which we believe has significantly strengthened our manuscript.

---

> > ### Comment · Reviewer_hXr5 · 2025-04-05
> >
> > Thank you for your response.
> >
> > ```
> > EEG-Text Alignment
> > ```
> >
> > While the inclusion of multimodal information appears to contribute to performance, this may be due to the additional, distinguishable information introduced by the word representations. Also, I am curious about the alignment between simulated and real clinical reports. How do you evaluate whether the simulated text generation aligns with real-world clinical EEG reports? Also, in many clinical settings, reports often include only brief notes, such as onset/offset times or summary-level observations, rather than rich detailed descriptions. Without evaluation, how do we assess the validity and fairness of the proposed method?
> >
> >
> > ```
> > Regarding the motivation for downstream tasks, we agree that EEG-only methods can address detection/classification. However, our multimodal approach enhances unimodal EEG encoder initialization, yielding better representations for these tasks.
> > ```
> > ```
> > The pretrained EEG encoders we are releasing can be used for downstream EEG-only clinical tasks without additional complexity and at reduced cost due to the low parameter count.
> > ```
> >
> > I think my main concern remains: why must we involve text data to improve EEG classification tasks, which are traditionally EEG-only?
> >
> > In clinical practice, doctors do not rely on text notes to perform detection or classification. For seizure diagnosis, standard detection is performed on EEGs. Some cases may require video monitoring, which is costly and only available in tertiary hospitals.
> > While you mentioned that "These reports, a byproduct of standard clinical workflows,...., clinical reports already available in large quantities in hospitals and with neglible costs to store compared to EEG data and especially compared to acquiring additional EEG data", my personal opinion is that this is not true. Generating suitable EEG clinical reports for model training is resource-intensive, and such data is generally limited to a small number of tertiary care centers. In fact, there are no large available EEG reports in general cases.  Many clinical EEG reports consist only of brief notes, lacking detailed descriptions. Some initiatives, such as TUH DB, stop providing clinical notes and instead encourage development focused on EEG detection methods. In general, there is no clear clinical motivation to incorporate text reports into EEG classification tasks.
> >
> >
> > ```
> > We believe that learning and evaluating pathology-sensitive representations is a critical first step
> > ```
> >
> > I think EEG classification is often used for phenotyping, including identifying stages, events, or abnormal activity, rather than explicitly detecting neuropathological patterns. Pathology tasks focus on more understanding underlying mechanism of diseases, like epileptogenic zone (EZ) or molecular-level information. It would be better to properly define the intended focus of the paper, and clarify what specific clinical or biological insights the text-based component contributes beyond performance improvements in classification accuracy.
> >
> > While the authors have clarified the potential of the proposal, my concerns remain. I am temporarily lowering my rating.

---

> > > ### Author Response · Authors · 2025-04-07
> > >
> > > Thank you for your continued feedback, which has been useful in refining our manuscript. We would like to address the misunderstandings that led to your lowered score. Below, we provide detailed clarifications.
> > >
> > > > I am curious about the alignment between simulated and real clinical reports.
> > >
> > > We apologize for any confusion: our method **does not generate or simulate clinical reports**. Instead, we use existing clinical reports from the TUEG dataset (a subset totalling 11.8K reports), naturally produced during standard practice. We expand on this below.
> > >
> > > > in many clinical settings, reports often include only brief notes, such as onset/offset times or summary-level observations, rather than rich detailed descriptions.
> > >
> > > We regret if “reports” suggested extensive documentation and apologize for the confusion. In reality:
> > > - **Brief Notes Are Common**: A portion of TUEG reports are short, yet effective. Reports do not need to provide highly detailed descriptions. Compared to binary labels, a brief note with observed EEG events and the clinical correlation already provides a much richer signal. In addition, this enables considerably more data, given our subset of 11.8K reports compared to the largest abnormal corpus (TUAB) with 2.7K labels.  A new figure shows heterogeneous report lengths, reflecting real-world diversity [Figure S6; [link](https://docs.google.com/document/d/e/2PACX-1vQygdcAED1qMhVgFv5jU9TsclAyRxp-XKFiGwxK2pkxLSdrKAgyGVuAEYBVPnmQZeJDfIBVMLTbzTwG/pub)].
> > >
> > > - **Scalability**: Given a clarification on length, we hope the reviewer agrees that similar notes exist more commonly across settings, rather than just tertiary centers, making our approach more broadly applicable.
> > >
> > > > why must we involve text data to improve EEG classification tasks, which are traditionally EEG-only? In clinical practice, doctors do not rely on text notes to perform detection or classification.
> > >
> > > We agree and would like to stress that **no text data is involved during downstream tasks**. To clarify,
> > >
> > > - **"EEG-Only methods"**: EEG representations are learned by only pretraining on EEG. These representations are subsequently evaluated on downstream tasks.
> > > - **Our EEG-Language methods**:  EEG representations are pretrained by aligning EEG signals with text embeddings from clinical reports, enriching the learned features with contextual guidance. For classification, the text encoder is discarded, and we evaluate the EEG representations alone. For zero-shot classification, a simple prompt (e.g., ‘EEG is abnormal/normal’) is embedded using the text encoder, but no patient-specific report/text is used or generated.
> > >
> > > Our results show this offers two key benefits: (1) improved EEG representations for classification tasks, especially in low-data scenarios, and (2) zero-shot capabilities - all without altering clinical workflows that rely on EEG alone at test time. We apologize for any confusion in our original presentation. To address this, we propose the following addition to Section 4.1:
> > > - *We emphasize that “EEG-only” refers to pretraining without text, while ELMs use text solely during pretraining to guide EEG representation learning. At test time, neither method uses clinical reports, ensuring alignment with standard EEG-based clinical practice.*
> > >
> > > > I think EEG classification is often used for phenotyping, including identifying stages, events, or abnormal activity, rather than explicitly detecting neuropathological patterns.
> > >
> > > We agree with the reviewer's characterisation of the primary use of EEG. We apologize if our use of 'pathology detection' has caused confusion with respect to the intent of the paper. We aimed to ameliorate this by always using 'detection'. While we would like to kindly note that it is commonly used in the literature for our scope of clinical phenotyping (e.g. [1-3]) we understand the reviewer's concern and propose to rephrase the abstract:
> > > - L14: Multimodal language modeling has enabled breakthroughs for representation learning, yet remains unexplored in the realm of functional brain data for **clinical phenotyping**.
> > > - L22: Compared to EEG-only models, our multimodal models perform significantly better **across four clinical evaluations** and (...)
> > >
> > > And clarify at the start of the introduction:
> > > - L35: While EEG sees widespread clinical use **for the detection of pathology, by which we refer to broad clinical phenotyping such as disease classification and event detection**, (...)
> > >
> > > Alternatively, we are open to adjusting the title to use 'clinical phenotyping' instead of 'pathology detection', although we might need ICML chair input on whether this is permitted and might risk misaligning the paper with the relevant literature. We hope this addition at the start of the introduction sufficiently clarifies our scope.
> > >
> > > (We provide DOI due to character restrictions)
> > > [1] https://doi.org/10.3390/math11071619
> > > [2] https://doi.org/10.1016/j.compbiomed.2021.104434
> > > [3] 10.1109/JSAC.2020.3020654

---

### Official Review · Reviewer_Ln87 · 2025-03-21

**Overall Recommendation:** 5

**Summary:**

The manuscript describes EEG-Language (CLIP-like) pretraining on medical EEG recordings and the accompanying textual medical reports. They used a pretrained medical langauge model and a from-scratch-trained EEG encoder to map temporal crops of EEG and subsections of medical reports to the same latent space, with some methodolical adaptations for the fact that multiple EEG crops belong to the same recording and multiple subsections belong to the same medical report. Experiments are performed across multiple medical EEG datasets. Results show that this yields good pathology detection accuracy in zero-shot settings, whihc can be further improved via linear probing on subsets of the training data and linear probing also works for seizure detection and event classification.

**Claims And Evidence:**

The claims of a EEG-language-model pretraining that yields good pathology detection from no or few labels seem well-supported to me.

**Essential References Not Discussed:**

None that I am aware of

**Experimental Designs Or Analyses:**

See Methods and Evaluation Criteria

**Methods And Evaluation Criteria:**

The authors provide a sensible setup for the evaluation, multiple types of evaluation (retrieval, different types of downstream performance) on multiple datasets.

**Other Comments Or Suggestions:**

-

**Other Strengths And Weaknesses:**

Figures look clean, writing seems easy to read

**Questions For Authors:**

In p.3 $L_\textrm{orth}$ is $h_\textrm{e}$ normalized? I Assume so, because why otherwise is this correlation and not covariance?

**Relation To Broader Scientific Literature:**

EEG-Language pretraining with medical reports is an open area that seems to not have been tackled yet and for which any results are very valuable for the research community

**Theoretical Claims:**

None

---

> ### Author Rebuttal · Authors · 2025-03-31
>
> Dear Reviewer,
>
> Thank you sincerely for your positive feedback on our manuscript. We are grateful to hear about the value of our contribution to the research community, as well as the clarity of our manuscript.
>
> Regarding your question about $L_{orth}$, $h_e$ is indeed L2-normalized. We apologize for this omission in the paper. We have corrected the description and notation in Section 3.1.1 accordingly.
>
> Thank you for helping us improve our manuscript.

---

> > ### Comment · Reviewer_Ln87 · 2025-04-01
> >
> > Thank you for addressing the confusion regarding the L2-normalization. I have another question, the medical reports, are they still publicly available for TUH, can anyone obtain them?

---

> > > ### Author Response · Authors · 2025-04-03
> > >
> > > Thank you for your question. Whereas all EEG datasets used in our manuscript are publicly available, the reports were provided by the Neural Engineering Data Consortium at Temple University following a data sharing agreement. While they offer search for keywords or items of interest (https://isip.piconepress.com/projects/nedc/html/tuh_eeg/), the full reports are currently not publicly available due to privacy regulations. Our hope is that our work encourages the clinical and research communities to recognize the potential of these reports when combined with advancements in modern machine learning, paving the way for broader access in the near future, much like what we've seen with radiology reports.
> > >
> > > We look forward to releasing our pretrained models (alongside our code), which can be readily used for both finetuning and inference without access to reports. We believe this will be a valuable resource for researchers and practitioners alike.

---

### Decision · Program_Chairs · 2025-05-01

**Decision:**

Accept (poster)

**Comment:**

This paper presents a creative approach for pretraining EEG models by aligning them with medical text reports, leveraging a combination of time-series cropping, text segmentation, and multiple instance learning (MIL) variant of contrastive learning. The proposed EEG-Language Models (ELMs) demonstrate convincing performance in various downstream tasks, including retrieval, abnormality classification, and event classification, across multiple datasets.

The reviewers overall acknowledge an innovative approach, comprehensive experimental results, and the potential of multimodal pretraining in medical applications.

Although 1 reviewer out of 4 raises some concerns (e.g. the lack of clinical applicability of the method) the discussion during rebuttal period among reviewers did not affect the strong support for acceptance by the 3 positive reviewers.

I endorse this paper for publication at ICML, as it presents a creative and impactful approach that bridges ML research on text and neural signals with the potential to advance the state of the art in EEG analysis and medical applications.